# Supra-biological performance of immobilized enzymes enabled by chaperone-like specific non-covalent interactions

Héctor Sánchez-Morán [1], Joel L. Kaar [1] ✉ & Daniel K. Schwartz [1] ✉

Designing complex synthetic materials for enzyme immobilization could unlock the utility of biocatalysis in extreme environments. Inspired by biology, we investigate the use of random copolymer brushes as dynamic immobilization supports that enable supra-biological catalytic performance of immobilized enzymes. This is demonstrated by immobilizing *Bacillus subtilis* Lipase A on brushes doped with aromatic moieties, which can interact with the lipase through multiple non-covalent interactions. Incorporation of aromatic groups leads to a 50 °C increase in the optimal temperature of lipase, as well as a 50-fold enhancement in enzyme activity. Single-molecule FRET studies reveal that these supports act as biomimetic chaperones by promoting enzyme refolding and stabilizing the enzyme's folded and catalytically active state. This effect is diminished when aromatic residues are mutated out, suggesting the importance of π-stacking and π-cation interactions for stabilization. Our results underscore how unexplored enzyme-support interactions may enable uncharted opportunities for using enzymes in industrial biotransformations.

Immobilization of enzymes on solid supports has the potential to provide increased stability, activity, and reusability[1–4], for applications in chemical and pharmaceutical manufacturing[5–10], biosensing[11–13] and bioremediation[14,15]. Typical immobilization materials consist of highly porous carbohydrate-based materials, inorganic supports or composites, including agarose[16–18], inorganic nanoflowers[19,20] or metal-organic frameworks[3,21–23]. However, enzymes are often inactivated upon immobilization on such materials due to denaturation as a result of enzyme-material interactions and/or blockage of the enzyme's active site, which in turn leads to a reduction in overall catalytic performance[24–26]. Recent breakthroughs suggest the potential of complex heterogeneous materials —such as lipid bilayers[27,28] or copolymer brushes[25,29–32]— as bespoke supports for enzyme immobilization. These synthetic materials are exceptionally dynamic and chemically tunable, comprising novel chemical moieties that interact non-covalently via transient self-assembly with moieties on the enzyme surface. Importantly, polymer brush supports enable the introduction of biomimetic non-covalent interactions, like H-bonding, hydrophobic self-assembly, π-stacking, and π-cation interactions. The latter

aromatic-related interactions are found in biological macromolecular assemblies including double-stranded DNA[33], proteins from extremophilic organisms[34–36] and intracellular liquid-liquid phase separation milieus[37–39]. Such interactions are also believed to contribute to thermal stabilization given their insensitivity to temperature variations[40], and can be twice as strong as salt bridges[41,42]. The incorporation of aromatic groups into immobilization supports may promote stabilizing π-stacking[38,43] and π-cation[34,35,42,44–46] interactions with aromatic or cationic surface-exposed enzyme residues, respectively.

The introduction of biologically-inspired stabilizing interactions into dynamic and chemically heterogeneous polymeric materials may enable complex stabilization mechanisms for soluble enzyme-polymer assemblies or immobilized enzymes. For instance, synthetic approaches to mimic biologically relevant interactions have been explored using heteropolymer ensembles[47]. Moreover, polymer encapsulation of enzymes has resulted in their stabilization at high temperatures and co-solvent interfaces[48–51]. While complex enzyme-polymer interaction mechanisms can be challenging to disentangle using conventional characterization methods, a wide range of biophysical, biochemical,

[1]Department of Chemical and Biological Engineering, University of Colorado Boulder, Campus Box 596 Boulder, CO 80309, USA.
✉e-mail: joel.kaar@colorado.edu; daniel.schwartz@colorado.edu

and computational approaches have been employed. Additionally, single-molecule Förster resonance energy transfer (SM-FRET) microscopy has provided unique mechanistic insights for enzymes immobilized on polymeric supports while specifically demonstrating that dynamic supports with a properly tuned balance of enzyme-polymer interactions may improve the performance of immobilized enzymes by stabilizing the folded state and/or facilitating refolding. The latter effect is analogous to a "chaperone-like" mechanism, whereby the native conformation and thus function of the enzyme is rescued from a denatured state. Notably, this phenomenon has been reported for several natural[27,28,52,53] and synthetic[25,54] molecules and macromolecules like lipid bilayers or surface-grafted polymer brushes. These findings suggest that other variations in polymer structure may offer new opportunities to enhance immobilized enzyme performance.

We previously showed that immobilization of *Bacillus subtilis* Lipase A (LipA) on zwitterionic sulfobetaine methacrylate (SBMA) polymer brush supports promoted notable thermal stabilization of LipA[25,29]. Unlike many other lipase enzymes, LipA lacks an amphiphilic lid that covers the active site and thus, its binding cleft is exposed to the solvent[55,56]. We found that the enhanced performance of LipA was due to stabilizing hydrophilic interactions between LipA's surface, which is overall highly hydrophilic relative to other lipases, and SBMA[26,57]. Additionally, we showed that more hydrophobic lipases were better stabilized by less hydrophilic mixed polymer brushes containing SBMA and polyethylene glycol methacrylate. Here, we investigated the hypothesis that doping SBMA polymer brushes with various amounts of aromatic ethylene glycol phenyl ether methacrylate (EGPMA) monomer can dramatically enhance the activity and stability of immobilized LipA −which is rich in aromatic residues− at supra-biological temperatures.

In contrast with previous efforts to stabilize enzymes with random copolymer brushes, the presence of the aromatic EGPMA component may potentially enable π-stacking and/or π-cation interactions. To further isolate the relevance of aromatic interactions, the impact of mutating aromatic residues in LipA on its stabilization by the incorporation of EGPMA was studied. Additionally, SM-FRET was employed to study the conformational dynamics of immobilized LipA and obtain mechanistic insights into the complex unfolding and refolding dynamics within the brush layer. Finally, the generality of this approach was demonstrated using an additional unrelated enzyme, carbonic anhydrase. Overall, the combined macroscopic and molecular-scale observations underscore the importance of properly tuned interactions between aromatic moieties in the immobilization support and enzyme surface residues for preserving and rescuing the folded and active state of immobilized enzymes.

## Results and discussion
### Synthesis and characterization of SBMA/EGPMA brushes
We aimed to investigate whether we could leverage a broad range of non-covalent interactions based on the structural features of LipA to further increase its stabilization compared to previous approaches. Interestingly, LipA has an unusually high number of solvent-exposed aromatic residues, accounting for ∼7% of its surface. The aromatic patches found on LipA are less hydrophobic than patches formed by aliphatic residues, which are present in more hydrophobic lipases[57]. Furthermore, the high surface hydrophilicity of LipA is also due to the presence of numerous charged and hydrophilic residues (lysine, arginine, aspartate, glutamate). Based on these observations, we hypothesized that LipA may be further stabilized by introducing monomers with aromatic groups within SBMA brushes, which may specifically interact with aromatic (π-stacking) and cationic (π-cation) residues. These aromatic groups may also potentially promote other interactions with LipA, like hydrophobic interactions with the solvent exposed aliphatic moieties that surround LipA's active site, potentially

improving its overall stability and catalytic performance via these other mechanisms[58].

To test our hypothesis, we prepared surface-grafted random copolymer brush supports composed of SBMA and 0–10% molar of aromatic EGPMA (Fig. 1a) via atom transfer radical polymerization from the surface of silica nanospheres with a hydrodynamic diameter of 411 nm. In addition to SBMA and EGPMA, the brushes included 5% molar glycidyl methacrylate (GMA), which contains an epoxide group that can react covalently with numerous sites on the surface of enzymes, including the N-terminus, lysines, cysteines, tyrosines, and histidines. Based on analysis of pKa and solvent exposure, the most likely sites on LipA that can react with GMA include the N-terminal amine and histidines, including the C-terminal 6xhis tag (Supplementary Table 1). The high likelihood of reactivity of numerous sites enabled LipA to be covalently immobilized despite the protein resistant properties of SBMA-rich brushes[59]. Assuming equal reactivity of these methacrylate monomers that we have characterized in prior work[54], we estimated that the mean molecular volume of GMA in the brush layer was approximately 2.5% of the brush layer for all brush supports, suggesting that the number of attachments between LipA and the brush layer was relatively independent of brush composition. Incorporation of EGPMA was confirmed using diffuse reflectance Fourier-transform infrared spectroscopy, which revealed a systematic increase of the aromatic C-H stretching mode with increasing EGPMA content (Fig. 1b). Characterization of the polymer-modified particles by dynamic light scattering suggested that the solvated polymer corona was 73–143 nm thick (Fig. 1c). Lastly, we verified the wettability of all brushes grown on flat silica wafers and found similar static contact angles of ∼11° for all surfaces (Supplementary Fig. 1), consistent with their highly hydrated state[60].

### Mixed SBMA/EGPMA supports confer supra-biological LipA activity and stability
To assay the activity of immobilized LipA as a function of temperature and EGPMA content in the brush layer, LipA was covalently immobilized on 0–10% EGPMA supports, which resulted in loadings of 0.16–0.38 mg of enzyme per g of support (Supplementary Table 2). The stability of free and immobilized LipA was assayed by measuring the initial rate of hydrolysis of resorufin butyrate as a function of temperature between 20–90 °C. We observed that immobilization of LipA resulted in both the activation and stabilization of LipA at elevated temperatures for all polymer brush compositions. Interestingly, the activity of immobilized LipA at elevated temperatures depended strongly on the EGPMA fraction in the brush layer. This can be seen in Fig. 2a, which shows the temperature-dependent activity profiles of free and immobilized LipA. Additionally, the stabilizing effect on LipA increased with EGPMA content up to 5% and then decreased for supports containing 10% EGPMA. Specifically, immobilization of LipA on the 5% EGPMA support resulted in a dramatic catalytic enhancement of up to 50-fold compared to the optimal measured activity of free LipA (V_opt), along with an increase in the temperature of optimal activity (T_opt) from 40 °C to 90 °C (Supplementary Fig. 2). Moreover, while free LipA showed a pronounced loss of catalytic activity above 40 °C, LipA immobilized on a 5% EGPMA support exhibited increasing activity over the entire measured temperature range. Notably, the 10% EGPMA supports were less stabilizing than the pure SBMA supports (without EGPMA). This non-monotonic trend suggested a nuanced stabilizing mechanism associated with an optimal fraction of EGPMA. For ease of visualization, the differential performance (i.e., turnover) of immobilized LipA as a function of temperature from Fig. 2a was represented as a heatmap (Fig. 2b). This highlighted the monotonic temperature-dependent increase of LipA turnover when immobilized on 5% EGPMA supports, in contrast with the significantly lower activity on 0%, 1%, and 2% EGPMA supports, and the loss of activity above 70 °C on 10% EGPMA supports. To further illustrate the non-monotonic relationship

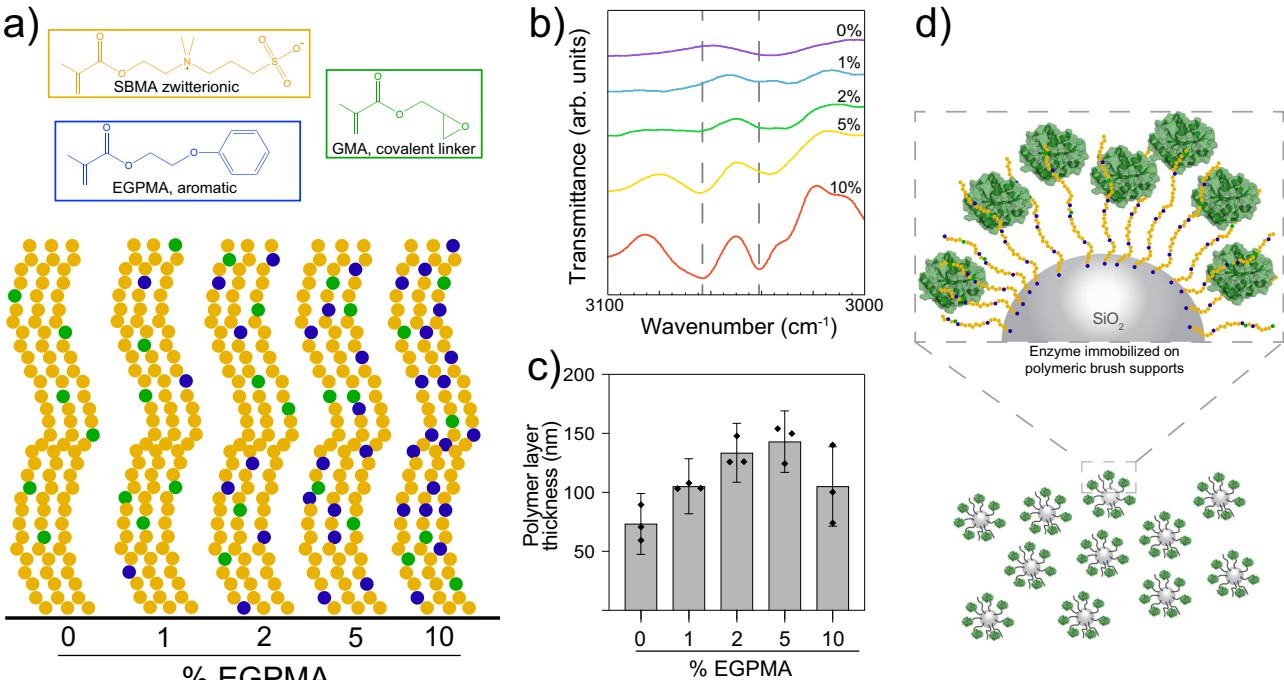

**Fig. 1 | Overview of immobilization of LipA on mixed SBMA/EGPMA random copolymer brushes. a** Schematic of the brush surfaces with chemical structures of SBMA, EGPMA, and GMA. **b** FTIR-DRIFTS spectra of 0−10% EGPMA polymer brushes grown on silica nanoparticles. Dashed lines represent characteristic peaks for aromatic C-H stretching modes. **c** Thickness of the hydrated polymer layer measured by dynamic light scattering. For each composition, the radius of the core SiO₂ particle was subtracted from the radius of the polymer-grafted particle. Error bars represent the standard error of the mean for three technical replicates ($n = 3$), accounting for propagation of error related to the subtraction of the core silica particle. **d** Illustration of the resulting brush supports after conjugation of LipA to GMA in the brush layer.

between LipA activity and brush composition, the turnover of the enzyme at 40 °C as a function of EGPMA content is shown in Fig. 2c. Notably, Supplementary Fig. 2 also shows the data from the temperature-dependent activity assays for soluble and immobilized LipA on a linear activity scale.

To better understand the underlying phenomena associated with the catalytic enhancement of immobilized LipA, we analyzed the kinetic parameters of immobilized LipA as a function of temperature. This analysis revealed that catalytic differences between support compositions at high temperatures could be attributed to differences in $k_{cat}$ and not $K_m$ (Supplementary Table S3). However, increases in $k_{cat}$ were insufficient to explain the entirety of LipA's catalytic enhancement. This suggested an interplay between an increase in $k_{cat}$ with temperature and other synergistic effects, such as increased active site preservation. Analysis of $K_m$ suggested that differences in substrate binding as a function of EGPMA content at constant temperature were negligible. Additionally, while $K_m$ values were higher at elevated temperatures for all brushes, indicating weaker substrate binding, this may be attributed to reduced absorption of the substrate to the brush layer as determined by measuring substrate partitioning (Supplementary Fig. S3). These findings altogether suggested that the apparent enhanced catalysis may indeed be explained by the thermal stabilization of LipA by the brush surfaces.

Previous work suggested that complex heterogeneous supports may prolong the operational lifetime of immobilized enzymes[25,27,29,61]. To study the ability of EGPMA-containing supports to preserve enzyme activity, we measured the time-dependent catalytic activity of immobilized LipA by incubation at 40 °C (Fig. 2d). Within one day, the decrease of relative activity was similar for all support compositions, although the 5% EGPMA support promoted a much higher initial LipA activity. However, the activity of immobilized LipA was retained for longer times by certain support compositions, namely 5% EGPMA. To

visualize the cumulative activity over time, we estimated the reaction yield of product formation by integrating the data in Fig. 2d, which represents the area under the activity retention curves. This analysis revealed that the 5% EGPMA support provided at least twice the specific product yield (Fig. 2e), which highlights the practical relevance of fine-tuning the EGPMA content of immobilization supports for stabilizing LipA at high temperatures.

### Removing aromatic interactions reduces thermal stability

Based on the stability results described above, we sought to mechanistically elucidate the effects of aromatic interactions on the stabilization of LipA. If such interactions contributed to LipA stabilization, the stabilizing effect of EGPMA should be reduced if aromatic residues in the enzyme were removed. This hypothesis was tested by designing three different LipA variants in which the relative amounts of surface-exposed and buried aromatic residues were mutated to alanine (exposed) or methionine/valine (buried). The mutants were designated LipA$_{\Delta SA-50\%}$, with solvent-accessible aromatic surface residues mutated to reduce the aromatic surface area by 50%, LipA$_{\Delta SA-100\%}$, with all surface aromatic residues removed, and LipA$_{\Delta buried-100\%}$, with all buried aromatic residues removed (Supplementary Table 4, Supplementary Fig. 4). These mutants allowed us to determine whether removing solvent-exposed aromatic residues accessible in the folded state (LipA$_{\Delta SA-50\%}$ and LipA$_{\Delta SA-100\%}$) resulted in reduced stabilizing π-stacking and π-cation interactions with EGPMA. Additionally, removal of buried aromatic residues accessible in the unfolded state (LipA$_{\Delta buried-100\%}$) allowed us to test whether reduced aromatic interactions with EGPMA in the unfolded state −which could hypothetically enhance refolding− might also lead to an overall decrease in activity and/or stability.

The mutants were immobilized on EGPMA-containing particle supports, with loadings in the range 0.14−2.28 mg of enzyme per g of

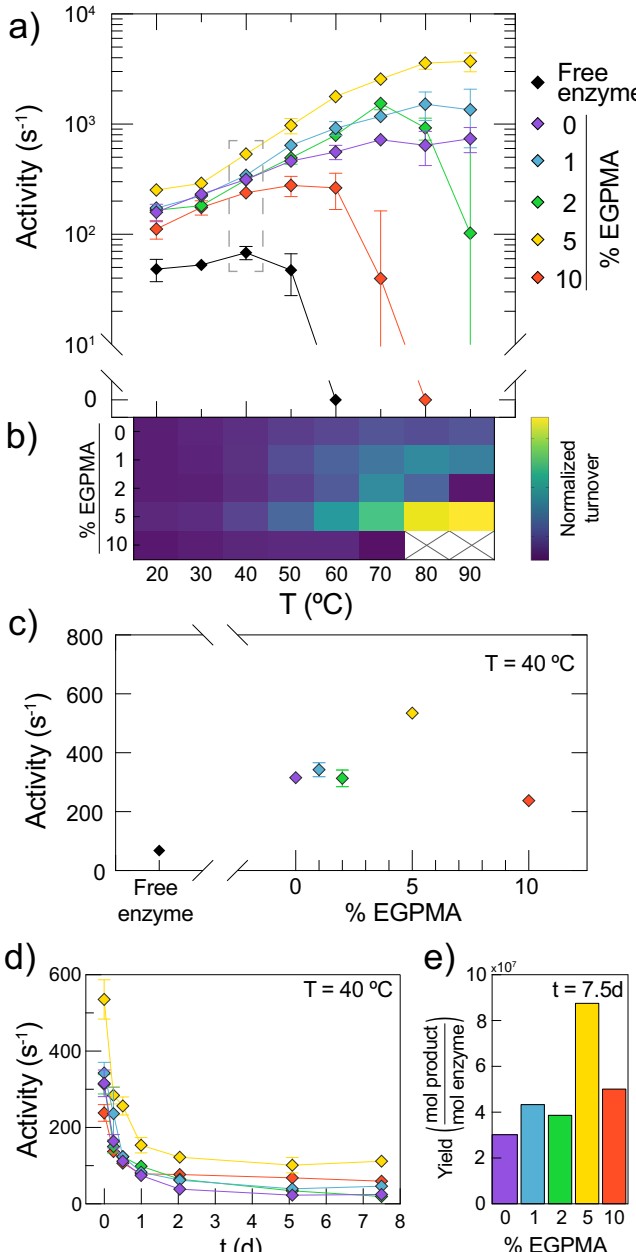

**Fig. 2 | Dramatic enhancements in LipA performance on mixed SBMA/EGPMA brushes. a** Temperature-dependent activity of immobilized and free LipA on SBMA/EGPMA supports. Error bars represent the standard error of the mean of three technical replicates for each measurement ($n = 3$). **b** Heatmap corresponding to the temperature-dependent activity data from panel **a** scaled to the interval between the lowest and highest turnover of immobilized enzyme. **c** Turnover of LipA at 40 °C obtained from data points in the dashed box of panel **a**. **d** Enzymatic activity retention profile of immobilized LipA during incubation at 40 °C. Error bars represent the standard error of the mean of three technical replicates for each measurement ($n = 3$). **e** Theoretical total yield of product formation corresponding to integration of curves from Fig. 2d. Error bars estimated via propagation of integrated error are included but are too small to be observed.

support (Supplementary Table 6), and their catalytic activities were measured. Heatmaps of activity revealed that LipA mutants with reduced aromatic content (exposed and buried) were stabilized on polymer supports with lower aromatic composition than wild-type LipA (LipA$_{WT}$), and that even optimized immobilization of these mutants was less stabilizing. Specifically, all three mutants were optimally stabilized on supports with less EGPMA (1% or 2%) than the

optimal 5% EGPMA support for LipA$_{WT}$ (Fig. 3). Additionally, all three immobilized mutants had reduced thermal stability relative to immobilized LipA$_{WT}$, losing activity above 70 °C, even though some of the soluble mutants were more active and/or stable than soluble LipA$_{WT}$. These findings suggested that the aromatic residues in LipA$_{WT}$ played an important role in promoting stabilizing interactions with EGPMA-containing supports, supporting the hypothesis that π-stacking and π-cation are involved in the stabilization of immobilized LipA$_{WT}$. Other potential mechanisms may also be pertinent, including the formation of hydrophobic niches within the brush layer that may promote hydrophobic interactions[62,63]. It is also plausible that changes in aromatic content may change the polymer chain rigidity[64], which could alter the self-assembly of the brush layer around the enzyme's folded and unfolded states. Additionally, to determine if the mutations in LipA impacted substrate binding, we measured the K$_m$ for the soluble form of each of the variants and showed that the values of K$_m$ for LipA$_{WT}$, LipA$_{\Delta SA-50\%}$, and LipA$_{\Delta buried-100\%}$ were similar whereas the K$_m$ for LipA$_{\Delta SA-100\%}$ was higher than the others. (Supplementary Table 5). The latter was not surprising since Y161A, one of the mutations in LipA$_{\Delta SA-100\%}$, is close to the active site and could affect substrate binding. Even with this difference, the differences K$_m$ are not sufficient to explain the differences in activity that were observed. Notably, the temperature dependent data in Fig. 3 is also shown plotted on a linear activity scale in Supplementary Fig. 5.

## Conformational states of immobilized enzyme determined using single molecule FRET

SM-FRET microscopy was used to investigate the mechanistic basis underpinning the influence of brush chemistry on immobilized LipA stability. In contrast with traditional ensemble and time-averaging metrics of protein stability, SM-FRET provides quantitative information on the conformational state dynamics of immobilized enzymes on a molecule-by-molecule basis. The SM-FRET experiments in this work combined highly multiplexed single-molecule localization with FRET, to measure conformational changes of individual enzyme molecules with nm-scale resolution[65,66].

For SM-FRET experiments, a previously reported LipA variant with a solvent exposed cysteine and the non-canonical amino acid p-azidophenylalanine was used[25]. This enabled the use of orthogonal click chemistry to label LipA site-specifically with FRET-paired Alexa Fluor 555 and Atto 647 N (LipA-FRET, Fig. 4a), which was confirmed by SDS-PAGE (Supplementary Fig. 6). The fluorophores were positioned such that they were distal from the active site and in close proximity (~2.4 nm) in the folded state of LipA relative to the Förster radius of R$_0$ = 4.59 nm (Supplementary Methods), resulting in high FRET. The donor and acceptor were also far apart in the primary sequence, resulting in large separations in the unfolded state. This separation was estimated by modeling the random coil unfolded state of LipA via a self-avoiding random walk simulation, yielding an average inter-fluorophore distance of approximately 9.8 nm (Supplementary Fig. 7). Thus, a large decrease in FRET was expected upon LipA-FRET unfolding. This was confirmed by subjecting free LipA-FRET to chemical denaturation, which resulted in a systematic decrease in FRET efficiency as a function of denaturant concentration, suggesting the disruption of the tertiary structure of LipA (Supplementary Fig. 8).

LipA-FRET was immobilized on brushes composed of 0%, 5%, and 10% EGPMA on imaging silica wafers, and SM-FRET experiments were performed at 20 °C and 45 °C, providing information about stabilization mechanisms under thermally stable and thermally denaturing conditions, respectively. At each temperature, a large number of trajectories (1300–3200; Supplementary Table 7) was collected to enable statistically rigorous measurement of both time-averaged quantities (e.g., average folded fraction) and dynamic parameters (e.g., apparent unfolding and refolding rates). Donor-acceptor histogram heatmaps were generated from all observations for each brush composition and

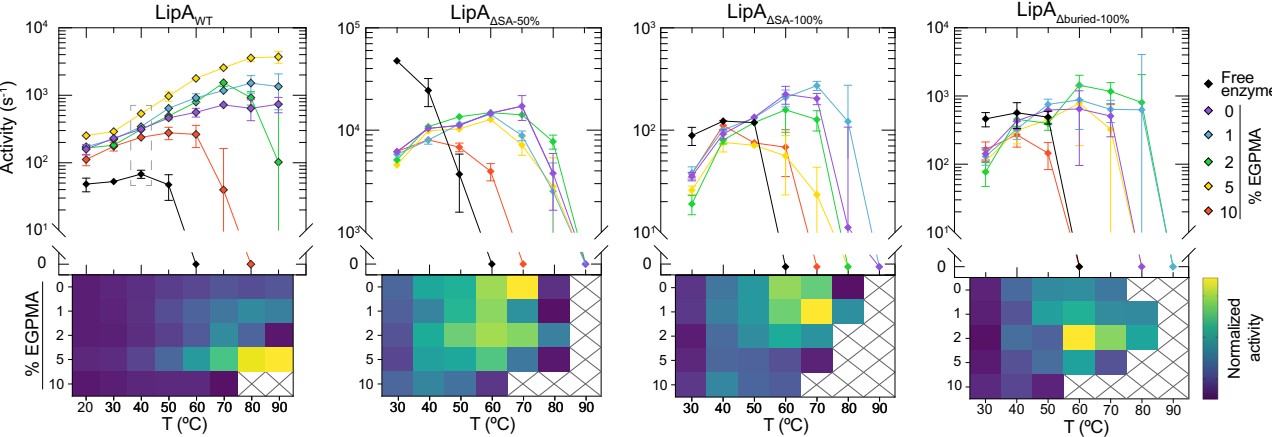

**Fig. 3 | Impact of mutations of surface exposed and buried aromatic residues on LipA activity on mixed SBMA/EGPMA brushes.** Activity as a function of EGPMA concentration is expressed as a plot and the corresponding heatmap for the various mutants as well as wild-type LipA. For each heatmap, color table scaling was based on the lowest and highest activity of the wild-type and mutant immobilized enzymes. Crossed rectangles correspond to conditions where immobilized and wild-type enzyme had no measurable activity. Note, the plot and heatmap for $LipA_{WT}$ are duplicated from Fig. 2a. Error bars for all plots represent the standard error of the mean of three technical replicates for each measurement ($n = 3$).

were used to identify folded and unfolded state populations for immobilized LipA in each experimental realization (Fig. 4b). In these 2D histograms, the population with low acceptor and high donor intensities corresponded to the apparent unfolded state, while the population with high acceptor and low donor intensities corresponded to the apparent folded state. For each brush composition, the populations were defined using a minimum integrated intensity threshold line as shown in Fig. 4b. The time-averaged folded fraction ($\phi_F$) for each brush composition was calculated as the number of folded-state observations divided by the total number of observations.

The analysis of folding fraction unexpectedly revealed that the 5% EGPMA support was uniquely capable of retaining folded LipA conformation under thermally denaturing conditions. Specifically, under non-denaturing conditions at 20 °C, immobilized LipA exhibited the highest folded fraction on the 0% EGPMA support (i.e., the SBMA homopolymer) with $\phi_F = 0.89 \pm 0.04$, while $\phi_F$ of LipA on 5% and 10% EGPMA was lower ($0.72 \pm 0.01$ and $0.66 \pm 0.04$, respectively). Notably, at 45 °C, the 5% EGPMA support preserved the same level of folding as at 20 °C ($\phi_F = 0.73 \pm 0.02$), while LipA-FRET immobilized on 0% and 10% EGPMA supports exhibited reduced stability ($\phi_F = 0.56 \pm 0.04$ and $0.60 \pm 0.02$, respectively).

These data suggested that all three support compositions conferred significant conformational stability under thermally denaturing conditions. Notably, 45 °C is above the $T_m$ for soluble LipA[67], and our activity measurements found that $T_{opt}$ of free LipA was ~40 °C; therefore, soluble LipA is largely unfolded at 45 °C. Thus, the non-zero values of $\phi_F$ at 45 °C for LipA immobilized on all three support compositions can be attributed to favorable brush-enzyme interactions and are consistent with the retention of activity by immobilized LipA shown in Fig. 2a. Because LipA is beyond its native $T_m$ at 45 °C, it was not surprising that $\phi_F$ was lower on the 0% and 10% EGPMA supports relative to at low temperatures. However, 5% EGPMA support was able to retain the same folded fraction at both low or denaturing temperatures, which illustrates the special nature of this support composition in stabilizing LipA. This in turn may explain the high activity and stability of LipA immobilized on this support as discussed above. Indeed, a qualitative correlation was observed between SM-FRET determined folded fraction and enzymatic activity at elevated temperature (Supplementary Fig. 9).

Interestingly, visual inspection of the FRET heatmaps revealed that the unfolded state population of LipA-FRET immobilized on the 5% EGPMA support was more compact (i.e., less diffuse) than the unfolded state population on 0% and 10% EGPMA supports. This suggested that the unfolded LipA-FRET immobilized on the 5% EGPMA support was more conformationally constrained compared to the other surfaces. This difference could also be observed in the distribution of donor intensities for the unfolded state on each surface, showing that the 5% EGPMA support indeed promoted a more constrained unfolded state (Supplementary Fig. 10).

## Compositionally tuned brushes reduce unfolding and accelerate refolding kinetics

To further understand the mechanisms that promoted a higher $\phi_F$ for LipA immobilized on 5% EGPMA supports in thermally denaturing conditions, dynamic information was extracted from the SM-FRET trajectories (Supplementary Fig. 11). Specifically, we determined transition probabilities between the folded and unfolded states for immobilized LipA-FRET, and used them to calculate the effective kinetic rate constants of unfolding and refolding ($k_{unfold}$ and $k_{fold}$) for each experimental condition (Fig. 5a) using a three-state Markov chain model. At 20 °C, the apparent decrease in $\phi_F$ with increasing EGPMA content correlated with a systematic decrease in refolding rate. Notably, at 20 °C, $k_{fold}$ was greatest for the 0% EGPMA brush ($6.54 \pm 2.10$ s$^{-1}$) and decreased systematically with increasing EGPMA content to 5% ($3.92 \pm 2.34$ s$^{-1}$) and 10% ($1.53 \pm 0.22$ s$^{-1}$). Simultaneously, $k_{unfold}$ was similar on 0% ($0.156 \pm 0.047$ s$^{-1}$) and 5% ($0.120 \pm 0.061$ s$^{-1}$) EGPMA brushes, and higher for the 10% EGPMA brush ($0.310 \pm 0.041$ s$^{-1}$).

An analogous analysis at 45 °C revealed a nuanced synergistic effect that promoted higher $\phi_F$ for immobilized LipA-FRET on the 5% EGPMA brush by both inhibiting unfolding and promoting refolding. Firstly, the 5% EGPMA brush promoted a higher $k_{fold}$ ($2.40 \pm 0.55$ s$^{-1}$) than the 0% ($2.13 \pm 0.29$ s$^{-1}$) and 10% ($1.11 \pm 0.17$ s$^{-1}$) EGPMA supports. Simultaneously, the 5% EGPMA brush inhibited the unfolding kinetics of immobilized LipA-FRET as evident by a lower $k_{unfold}$ ($0.298 \pm 0.063$ s$^{-1}$) compared to the 0% ($0.922 \pm 0.127$ s$^{-1}$) and 10% ($0.430 \pm 0.061$ s$^{-1}$) EGPMA brushes. Importantly, the interplay between the "chaperone-like" faster refolding and slower unfolding on 5% EGPMA brushes correlated with the apparent increase of $\phi_F$ that resulted in enhanced catalytic performance. To a first approximation, the behavior of LipA on the 5% EGPMA support can be explained by a stabilized folded state and a destabilized unfolded state.

Fluctuations in the folded and unfolded state were also measured by tracking the root-mean-squared fluctuations (RMSF) in inter-

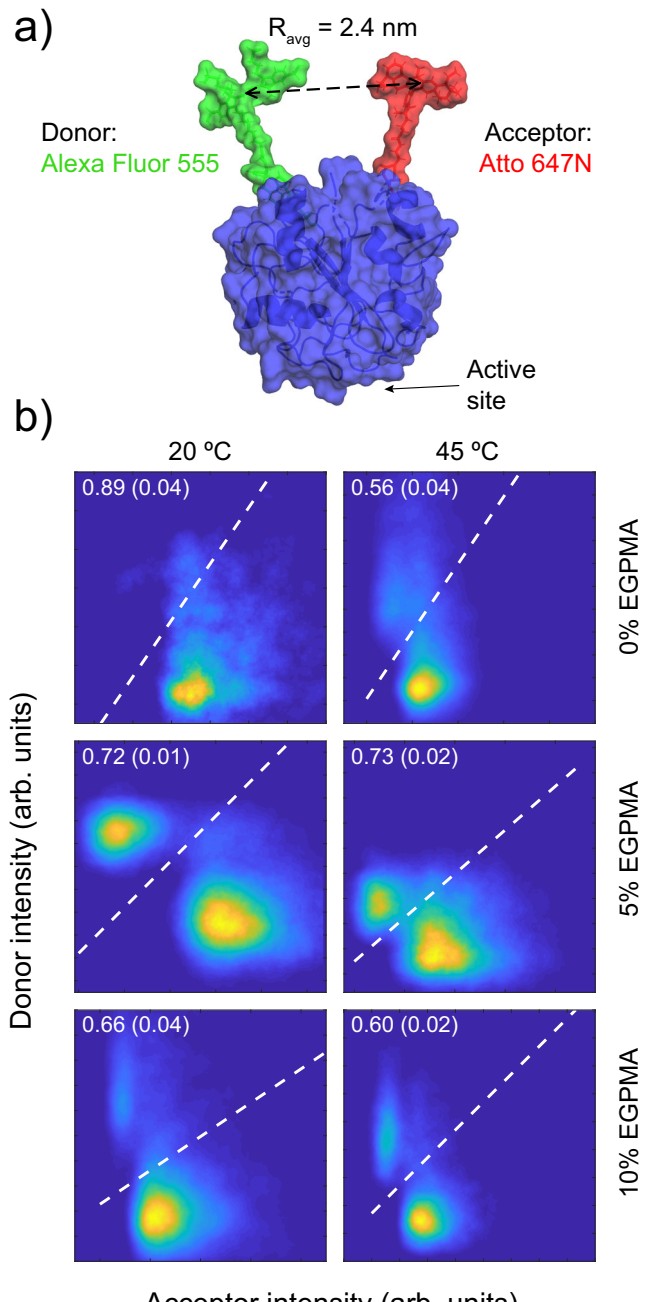

**Fig. 4 | LipA-FRET schematic and FRET heatmaps of immobilized LipA-FRET. a** Schematic of labeled LipA-FRET, with average inter-fluorophore distance ($R_{avg}$). **b** FRET heatmaps corresponding to each polymeric support composition (rows), and each temperature (columns). Dashed white lines indicate the threshold used to distinguish folded form unfolded observations. Annotated numbers represent the associated value of average folded fraction, $\phi_F$, for each condition. Numbers in parentheses represent the standard deviation of folded fraction for five trials ($n = 5$) of dashed line determination using different starting points prior to unsupervised minimization method.

fluorophore distance during intervals when the protein resided in one state or the other (Fig. 5b). Analysis of RMSF values of each state revealed nuanced dynamic trends with EGPMA content, which were qualitatively similar at both 20 °C and 45 °C. Specifically, the RMSF of the folded state of immobilized LipA decreased systematically with increasing EGPMA content, suggesting that increased protein-polymer aromatic interactions may restrict protein motion. This was consistent with the reduced $k_{unfold}$ observed on 5% and 10% EGPMA brushes with

respect to 0% EGPMA. The fluctuations of the unfolded state varied non-monotonically with EGPMA concentration, with reduced RMSF for the unfolded state on the 5% EGPMA brush. This trend was reminiscent of the aforementioned observations from Fig. 3b and Supplementary Fig. 10 consistent with the presence of a localized unfolded state. The 5% EGPMA brush apparently promotes a unique unfolded state that is less dynamic and more conformationally constrained; this unfolded state refolds at a faster effective rate compared to other brush compositions that promote less restricted unfolded states.

Analysis of the transition rate constants and fluctuations revealed a crucial role of EGPMA's aromaticity in the behavior of folded and unfolded LipA. Specifically, we hypothesized that monomer-residue π-stacking and π-cation interactions may occur between EGPMA phenyl moieties and LipA's numerous solvent exposed tyrosines, phenylalanines, and tryptophans (π-stacking); or lysines and arginines (π-cation), influencing the dynamics and stabilization of catalytically active LipA. For example, the folded state of LipA may be promoted by effectively lowering the free energy of the folded state via stabilizing aromatic interactions, decreasing $k_{unfold}$. Conversely, when unfolded, previously buried phenylalanines or tyrosines could form new π-stacking interactions with EGPMA as well as π-cation interactions with the quaternary ammoniums of SBMA. Since the timescale of polymer side chain fluctuations is orders of magnitude faster than enzyme unfolding[68,69], the brushes could probe the newly exposed residues and self-assemble around previously buried aromatics. We speculate that such interactions may facilitate refolding by effectively trapping the enzyme in a partially unfolded state along the unfolding process, resulting in the localized and dynamically restricted unfolded state observed for 5% EGPMA supports. This in turn may reduce the probability of LipA accessing a slow or kinetically trapped unfolded state. Thus, by inhibiting the unfolding process, the optimally tuned 5% EGPMA support may help funnel LipA back to its folded, catalytically active conformation, which was consistent with the accelerated $k_{fold}$ relative to other polymer compositions.

## Stabilization of carbonic anhydrase on EGPMA-containing brushes

To determine whether brushes containing EGPMA can be used to stabilize an enzyme that is different from LipA in structure and function, we selected a thermostable mutant of human carbonic anhydrase II (hCAII)[70]. Carbonic anhydrase has potential use in carbon capture applications, as it can sequester carbon dioxide via hydration into carbonic acid[71,72]. We immobilized hCAII on EGPMA-containing brushes with loadings in the range 0.81–8.60 mg of enzyme per g of support (Supplementary Table 8), and assayed the enzymatic activity for free and immobilized enzyme as a function of temperature (Fig. 6 and Supplementary Fig. 12), using 4-methylumbelliferyl acetate as a fluorogenic substrate. While free enzyme was more active than immobilized at low temperature, our results suggested that all supports increased the thermostability of hCAII, suggesting a trade-off between activity and stability for hCAII. Moreover, thermal stabilization increased systematically with EGPMA content, with 10% EGPMA support being notably more stabilizing than other compositions, achieving an optimal activity 9-fold higher than free enzyme (Supplementary Fig. 12). Importantly, all supports containing EGPMA surpassed free enzyme in catalytic performance at elevated temperatures, whereas only the support that did not contain EGPMA failed to outperform free hCAII in activity. These observations demonstrate the relevance of EGPMA supports for promoting hCAII stabilization and increased activity.

Examining hCAII's structural features, we found it contains many residues that may favorably interact with aromatic moieties of EGPMA (Supplementary Fig. 13). For example, hCAII contains 23 solvent-exposed lysine and 7 arginine residues, all of which can participate in π-cation interactions. Additionally, in contrast with LipA, which has

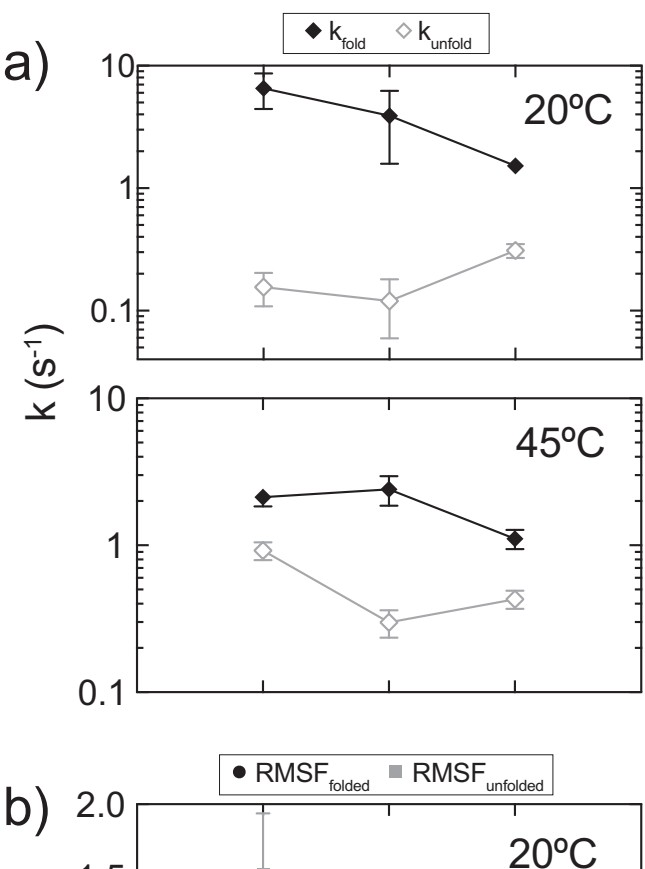

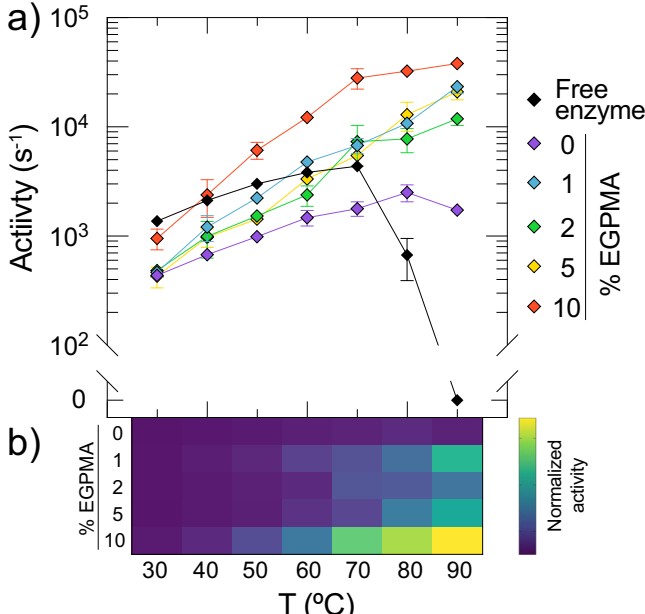

**Fig. 6 | Stabilization of hCAII on mixed SBMA/EGPMA brushes. a** Temperature-dependent activity of immobilized and free hCAII on SBMA/EGPMA supports. Error bars represent the standard error of the mean of three technical replicates for each measurement ($n = 3$). **b** Heatmap corresponding to the temperature-dependent activity data from panel **a** scaled to the interval between the lowest and highest turnover of immobilized enzyme.

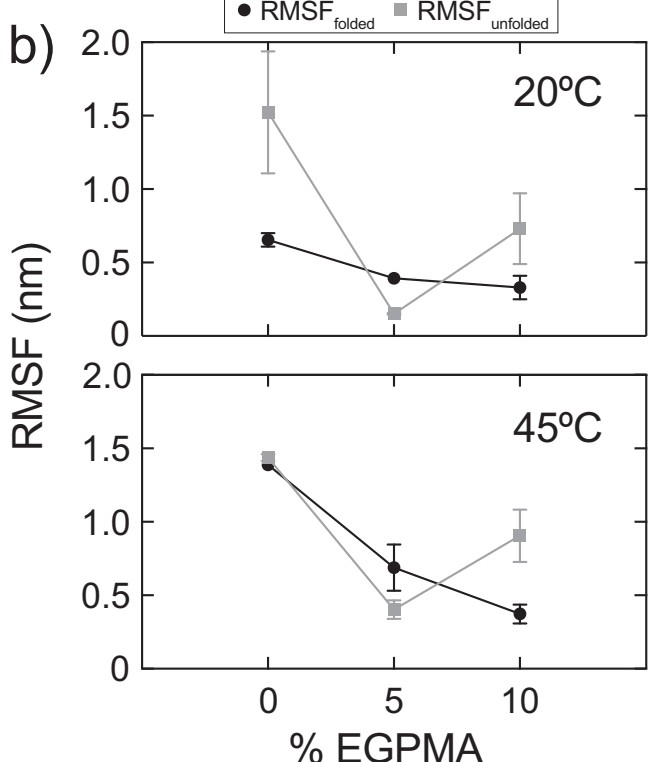

**Fig. 5 | Dynamic analysis of LipA-FRET immobilized on SBMA/EGPMA brushes. a** Effective transition rate constants for folding and unfolding for each studied condition. Error bars represent the standard error of the of rate constants, derived from the square root of the Cramèr–Rao lower bound as described by Kienle et al.[80]. **b** Fluctuations of the folded and unfolded states measured by the root-mean-square fluctuations (RMSF) of individual trajectories in intervals when protein remains in the folded or unfolded states, respectively. Error bars were estimated as the standard error of the means for ten iterations ($n = 10$) of jackknife resampling of the RMSF measurements for each condition.

more solvent-exposed than buried aromatics, hCAII contained 11 solvent-exposed and 16 buried aromatic residues which may be involved in π-stacking with EGPMA moieties. Based on the findings from hCAII activity and its structural features, we can conclude that immobilization of hCAII on EGPMA-containing supports greatly improves hCAII thermal stability and activity, likely due to multiple π-cation and π-stacking interactions between hCAII and the brush layer. Overall, these results generalize our observations suggesting that EGPMA supports are remarkably stabilizing for enzymes that are rich in aromatic and cationic residues.

Polymer brush supports comprising zwitterionic/aromatic (SBMA/EGPMA) random copolymers significantly enhanced the catalytic performance and thermal stability of immobilized LipA, an enzyme with high aromatic content. Specifically, an optimally tuned 5% EGPMA support promoted a 50 °C increase in $T_{opt}$ and a 50-fold increase in $V_{opt}$ compared to free enzyme. LipA mutants with reduced exposed or buried aromatic content were stabilized to a lesser extent, suggesting that stabilization may involve π-stacking and/or π-cation interactions involving the aromatic groups in the brush. SM-FRET analysis showed that 5% EGPMA brushes more effectively preserved the folded state of immobilized LipA against thermal denaturation compared to other SBMA/EGPMA supports. Stabilization of the folded state was consistent with reduced unfolding kinetics for immobilized LipA on 5% EGPMA brushes. Enhanced refolding kinetics of transiently unfolded enzyme as well as SM-FRET fluctuation analysis suggested that immobilized LipA on 5% EGPMA brushes exhibited a highly constrained unfolded state that refolded readily compared to a typical random coil, which may explain the enhanced refolding kinetics. These findings illustrate the potential of complex materials to impart supra-biological properties to enzymes while also opening opportunities to explore new heterogeneous polymers. Future work will explore further enzyme stabilization in increasingly complex supports, including multi-component mixtures that also contain charged, hydrophobic, H-bonding donors/acceptors, or strong dipoles. Ultimately, these new materials will provide unprecedented opportunities to allow the use of

enzymes in harsh environments and thus catalyze exceedingly difficult and interesting synthetic biotransformations. By promoting the stabilization of enzymes in such environments, our findings may furthermore have significant impact on bioprocess optimization and biosystems reaction engineering.

## Methods

### Materials
Copper (II) bromide, ascorbic acid, N,N,N′,N′′,N′′-pentamethyldiethylenetriamine (PMDETA), glycidyl methacrylate (GMA), ethylene glycol phenyl ether methacrylate (EGPMA), tetraethyl orthosilicate, ammonium hydroxide (30% in water), toluene, bicine and tris(2-carboxyethyl)phosphine (TCEP) were purchased from Millipore Sigma (St. Louis, MO). [2-methacryloyloxy)ethyl]dimethyl-(3-sulfopropyl)-ammonium hydroxide) (SBMA) was purchased from TCI Chemicals (Portland, OR). Methanol (MeOH) was purchased from Fisher Scientific. Dimethyl formamide (DMF) was purchased from Macron Fine Chemicals (Center Valley, PA). (p-chloromethyl)phenyl trichlorosilane (CMPS) was purchased from Gelest (Morrisville, PA). All chemicals were used without further purification.

Two-inch fused silica wafers for single molecule imaging were purchased from Mark Optics (Santa Ana, CA). Two-inch silicon wafers with native oxide coating for ellipsometry were purchased from Wafer Pro (Lafayette, CO).

Invitrogen Alexa Fluor 555 Click-iT strained dibenzyl cyclooctyne (AF555-sDIBO) was purchased from Thermo Fisher (Waltham, WA). ATTO-TEC Atto 647N-maleimide was purchased from Millipore Sigma (St. Louis, MO).

### Surface preparation, functionalization, and characterization
Silicon oxide nanospheres were synthesized following the Stöber method[26]. 75 mL of tetraethyl orthosilicate was dissolved in 1.536 L of a 1:1 methanol/ethanol mixture. Subsequently, 389 mL of ammonium hydroxide (30% in water) was added dropwise during 5 minutes under vigorous stirring and kept at room temperature for 4 h since the addition of the last drop. Particles were separated via centrifugation at $12,000 \times g$ for 5 min and washed with water (x2), ethanol, and methanol before storing them in a desiccator until further use. Subsequently, nanospheres were functionalized with CMPS via liquid deposition in anhydrous toluene, using CMPS in 0.1% v/v for 45 min. After deposition was completed, wafers were rinsed with toluene (x2), isopropanol (x2) and water, and stored in a vacuum desiccator until further use.

Surface-initiated polymerizations of mixed SBMA/EGPMA brushes were synthesized by the Activators ReGenerated by Electron Transfer Atom Transfer Radical Polymerization (ARGET ATRP) method[25,73–75]. Briefly, 2.5 g of CMPS-functionalized particles were added to 20 mL of a mixture of 7:3 (v/v) MeOH/DMF with SBMA/EGPMA in appropriate ratios, 5% glycidyl methacrylate (42.1 μL), 11.2 mg of L-ascorbic acid, and a small copper wire. A separate stock solution with copper and ATRP ligand was made separately, containing 0.0426 g of copper (II) bromide, 398 μL of PMDETA, 210 mL of methanol and 90 mL of DMF. All solutions were degassed in Schlenk flasks by repeated freeze-pump-thaw cycles (4x). Monomer-containing solution had mixtures of SBMA and EGPMA in ratios 100:0, 99:1, 98:2, 95:5 and 90:10. The copper and ligand-containing solution (10 mL) was transferred to the flask containing the nanospheres and monomer mixture under nitrogen to a final volume of 30 mL, which was allowed to react for 3 h at a positive nitrogen pressure of 5 psi. The final concentration of copper was 15 ppm, and molar ratios of all other chemicals are summarized in Supplementary Table 9. Polymer brush-coated nanospheres were separated via centrifugation, sequentially washed with methanol, DMF, water, and methanol, and stored in a desiccator until further use. The resulting particles were characterized via FTIR using a Thermo Fisher Scientific Nicolet 6700 FT-IR instrument with a DRIFTS sample holder. Dynamic light scattering measurements (Anton Paar Litesizer 500)

were performed after resuspension in 0.1 μm filtered 50 mM sodium phosphate buffer at pH 7.5 and probe sonication for particle dispersion.

For single molecule microscopy, silicon oxide wafers were cleaned in piranha solution (30% v/v hydrogen peroxide +70% sulfuric acid) for 1 h, rinsed with ultrapure water, dried with nitrogen, and exposed to UV-ozone for 1 h. After CMPS deposition, polymerizations were performed similarly as described for particles, but in a chamber with an inert $N_2$ atmosphere. Wettability of surfaces was measured by contact angle goniometry via the sessile drop method with a custom-built goniometer.

### LipA mutants design and expression
Mutants with different degrees of aromatic residue removal (LipA$_{\Delta SA-50\%}$, LipA$_{\Delta SA-100\%}$ and LipA$_{\Delta buried-100\%}$) were designed using Rosetta (Supplementary Table 4). Surface-accessible aromatic residues were replaced by alanine mutations (Supplementary Fig. 14), and Rosetta Energy Score was used to verify that introduction of each mutation did not introduce major disturbance of the folded state. For replacement of buried aromatic residues, Rosetta was allowed to select the most energetically favorable mutation. Buried Y130 was not mutated since its removal resulted in a dramatic decrease of the Rosetta Energy Score, suggesting that it was a critically stabilizing residue, and no non-aromatic replacement could compensate for its removal. After Rosetta-guided mutant design, structures were allowed to relax using the Rosetta relaxation protocol. Once mutations were selected, the genes for these mutants were introduced into pET21-b(+) expression vectors by Genscript via codon optimization. The genes of all mutants are shown in Supplementary Table 10.

### Enzyme expression, purification and labeling
The FRET construct of *Bacillus subtilis* Lipase A was expressed recombinantly with a modified version of a previously described autoinduction protocol[51]. Plasmids pET21-b(+) encoding LipA with a C-terminal hexahistidine tag and pDule2 pCNF RS encoding a tRNA synthetase for incorporation of the unnatural amino acid p-azidophenylalanine (pAzF) at an amber stop codon were co-transformed into *E. Coli* BL21(DE3). LipA contained two mutations at positions 4 and 175 for incorporation of pAzF and cysteine, respectively, to label LipA site-specifically with FRET-paired fluorophores via click reactions. Colonies were selected from a LB agar plate containing 100 μg/mL of ampicillin and 50 μg/mL of spectinomycin. After overnight growth up to saturation in non-inducing standard LB medium supplemented with the aforementioned antibiotics, cells were centrifuged, supernatant was removed, cells were resuspended in sterilized ultrapure water and transferred to 2 L culture flasks containing autoinduction minimal growth medium with antibiotics[76]. Additionally, sterile-filtered L-arabinose was added to a final concentration of 3.33 mM. Incubation was carried out for 27 h at 37 °C, with rotation at 250 rpm and flasks protected from light in aluminum foil. Following expression, cells were harvested via centrifugation at $5000 \times g$ at 4 °C during 30 min and subsequently flash frozen until purification.

Purification via affinity chromatography was carried out with a Ni-NTA column (Bio-Rad Laboratories). Briefly, cells were resuspended in buffer A, containing 50 mM sodium phosphate, pH 7, 500 mM NaCl, 10 mM imidazole and 6 M guanidine hydrochloride, and lysed via sonication. Cell lysate and subsequent samples containing LipA were protected from light in all cases. After centrifugation to remove cell debris, cell lysate was loaded on the Ni-NTA column and eluted with buffer B, which consists of buffer A and 300 mM imidazole. All lines, column and retrieved fractions of the FPLC system were protected with aluminum foil to prevent light-induced degradation of the aryl-azide group. Fractions containing LipA were identified using a Nanodrop spectrophotometer instead of using the 280 nm lamp of the chromatography system. Subsequently, LipA was buffer exchanged

into 50 mM sodium phosphate at pH 7 with 6 M guanidine hydrochloride using a Bio-Scale Mini Bio-Gel 50 mL P-6 desalting cartridge (Bio-Rad, USA). Final purity and concentration were assessed via SDS-PAGE and spectrophotometry, using an extinction coefficient of $22{,}900\ M^{-1}cm^{-1}$.

Labeling was carried out with a modified version of previously described methods[25,77]. 400 µL of 35 µM LipA were incubated with 0.1 µL of 2-mercaptoethanol and 0.3 mM DIBO-AF555 and allowed to rotate for 12 h at room temperature. Following AF555 labeling, LipA-AF555 was desalted 4 times into 50 mM sodium phosphate at pH 7 with 6 M guanidine hydrochloride buffer using 7 kDa Zeba spin columns (Thermo Fisher) to remove the presence of 2-mercaptoethanol, which could interfere with the next step.

Following buffer exchange, TCEP and Atto647N-maleimide –in that order– were added to final concentrations of 1.4 mM and 0.3 mM, respectively. After 6 h of rotation at room temperature, LipA-FRET was desalted 5 times with 7 kDa Zeba spin columns to remove excess free fluorophore. Removal of free fluorophore was assessed via nanodrop UV-Vis by stabilization of 553/280 and 648/280 signal ratios. Afterwards, labeling efficiency of LipA-FRET was assessed by measuring absorbances at 280, 553 and 648 nm, using extinction coefficients of $1.55 \times 10^{5}\ M^{-1}cm^{-1}$ $1.5 \times 10^{5}\ M^{-1}cm^{-1}$ for AF555 and Atto647N, respectively, while applying correction factors for overlapping regions (Supplementary Fig. 15). Labeling efficiencies were estimated as 48.1% for AF555 and 67.2% for Atto647N.

Expression of wild-type LipA and mutants was carried out in *E. Coli* BL21(DE3). The pET21-b(+) vectors that encoded each gene were transformed and single colonies were selected using 100 µg/mL of ampicillin as antibiotic. Expression of each gene was induced 1 mM of isopropyl β-D-1-thiogalactopyranoside and allowed to express at 26 °C for 24 h with rotation at 200 rpm. Harvesting and purification were carried identical to LipA-FRET.

Thermostable hCAII was kindly provided by Ryan Mehl (Oregon State University) encoded in a pET28b+ vector, and expressed in *Escherichia coli* BL21(DE3) as described previously[70,78].

## Enzyme immobilization and activity assays

To enable immobilization, purified enzymes (LipA wild-type, mutants or hCAII) were buffer exchanged into an immobilization buffer consisting of 10 mM bicine, pH 8.7, 100 mM NaCl. A solution of each enzyme (1.8 mL) was incubated with the polymer brush-coated nanospheres (200 mg) using a revolving orbital shaker. All enzymes were incubated for 12 h at 4 °C. After immobilization, enzyme loading on the particles was determined using a mass balance based on the relative activity of each enzyme remaining in the supernatant of the reaction. Relative activity was calculated by normalizing the activity of the enzyme in the supernatant to the activity of the enzyme in the absence of particles. To ensure that all immobilized enzymes were bound to particles via covalent attachment, we removed any non-covalently adsorbed enzyme from the polymer brush layer by extensively washing the particles with fresh buffer consisting of 50 mM sodium phosphate at pH 7 until the enzymatic activity of the supernatant was negligible.

Experiments of relative activity retention were carried out using a dry heating bath for incubation of enzyme-containing particles at 40 °C for 7 days, and a Tecan Infinite Pro 200 plate reader for fluorescence measurements, using excitation and emission wavelengths of 530 and 593, respectively. For each time point of this experiment, enzymatic activity was acquired by resuspending enzyme and pipetting out a constant amount in a well containing fresh buffer (50 mM sodium phosphate buffer at pH 7) with resorufin butyrate (final concentration of 50 µM). In all cases, non-specific substrate consumption was subtracted from the observed rate.

All temperature-dependent activity profiles were measured using a Horiba Scientific FluoroMax-4 spectrofluorometer with an incorporated Peltier temperature controller. The concentration of free and immobilized enzyme in each assay reaction, which was mixed continuously, was normalized based on the enzyme loading on the microspheres. Reactions were carried out in 50 mM sodium phosphate at pH 7, using 50 µM resorufin butyrate as fluorogenic substrate, and the rate of fluorescent product release was tracked in 0.1 s steps for 30 s after addition of substrate. A calibration of fluorescence units versus free resorufin concentration was used to convert the slope to units of product concentration (Supplementary Fig. 16), using excitation and emission wavelengths of 570 and 593 nm, respectively, using slit widths of 1 nm for both excitation and emission. Additionally, the background substrate hydrolysis was measured, subtracted from activity measurements at each temperature and verified that fluorescence signal was stable across the 20–90 °C temperature range studied (Supplementary Fig. 17). For measuring temperature-dependent activity profiles, the buffer was preheated to the target temperature in an external water bath. We also estimated potential resorufin butyrate partitioning in supports (Supplementary Fig. 3) by incubating supports without GMA or enzyme with 50 µM resorufin for 20 min at room temperature to enable resorufin adsorption in all supports and measure differences as a function of EGPMA content. After incubation, we split each sample in two equal parts. One part was washed three times with 50 mM sodium phosphate buffer at pH 7 and at 20 °C. The other part was washed three times with the same buffer at 70 °C. After these steps, we measured the remaining amount of resorufin in particles that had been treated for both temperatures, via fluorescence measurement of particles suspended in continuously stirred buffer in the spectrofluorometer.

For hCAII assays, we used 4-methylumbelliferyl acetate at final concentration of 100 µM as substrate. Similar to resorufin butyrate, we used a fluorescence calibration curve for the product 4-methylumbelliferone (Supplementary Fig. 18), and measured temperature-dependent hydrolysis and temperature-dependent fluorescence of substrate and product, respectively (Supplementary Fig. 19).

## Single molecule imaging

Prior to imaging, all buffers were photobleached overnight using a 100 mW near-UV laser. The polymer-functionalized imaging wafer was transferred into a custom-built flow cell with a window for imaging, which was initially filled with immobilization buffer. Separately, LipA-FRET, which was initially stored in a 50 mM sodium phosphate buffer with 6 M guanidine hydrochloride, was diluted into immobilization buffer (10 mM bicine, pH 8.7, 100 mM NaCl) to a final concentration of $5 \times 10^{-12}\,M$. Additionally, unlabeled wild-type LipA, was added to the same solution in a 1000 excess with respect to LipA-FRET, and all were introduced into the flow cell. After incubation for 2 h to allow immobilization to occur, the flow cell was washed with 30 mL of buffer (10 mM sodium phosphate, pH 7) and allowed to equilibrate for 30 min. Samples were imaged using a custom prism Total Internal Reflection Fluorescence (TIRF) apparatus using a Nikon Eclipse TE-2000 microscope with a 60× water immersion objective. The donor AF555 molecules were excited by 532 nm laser (Cobolt, Samba) and a 50 mW 640 nm laser (Crystalaser). The laser was directed into the back of the fused silica support through a fused silica hemicylindrical prism of 12.7 mm radius (Egorov Scientific) using a fused silica refractive index-matching oil. The emission of both AF555 donor and Atto 647 N acceptor fluorophores from LipA-FRET were separated into spectrally distinct imaging channels using a 610 nm dichroic mirror (T610LPXR, Chroma). The donor signal was further cleaned using a 585/29 nm band pass filter (Semrock), and the acceptor signal was cleaned with a 685/40 nm band pass filter (Semrock). Donor and acceptor imaging channels were aligned using an Optosplit III beam splitter (Cairn Research) and captured using an Andor iXon3 888 EMCCD camera maintained at −95 °C. Image acquisition was obtained using alternating

excitation with 200 ms of acquisition time. Thirteen movies were collected for each condition (temperature and/or brush composition) over different regions of the surface. After post processing using custom MATLAB software as described in previous work[29,79], 1350–3150 trajectories were included in the analysis for all conditions (Supplementary Table 7). Trajectories with abnormally high or overlapped fluorescent signals (LipA-FRET aggregates) or non-dual-labeled objects were excluded from further analysis. All trajectories of 4 or more frames were analyzed for 1 minute or until donor or acceptor fluorophores photobleached. FRET heatmaps were generated using all data points from each trajectory, kinetic transition parameters were estimated using a Hidden Markov Chain with three states as previously described[25,80]. Fluctuation analysis was carried out by estimating interfluorophore distance based on donor and acceptor intensities. Each root-square-mean fluctuation (RMSF) calculation was performed for all data points that corresponded a given folding state (folded or unfolded). If a protein molecule made multiple transitions between folded and unfolded states, all dwelling intervals in each state were considered as separate fluctuation events for RMSF calculation.

### Reporting summary
Further information on research design is available in the Nature Portfolio Reporting Summary linked to this article.

## Data availability
The experiment data that support the findings of this study are available upon request. Source data are provided with this paper.

## Code availability
The code used to analyze the data that support the findings of this study are available upon request. The Hi-patch code used for surface-exposed aromatic analysis can be found in Github: https://github.com/KaarLab/hi-patch.

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

## Acknowledgements

This work was supported by the NSF award 2103647 to J.K.L, and D.K.S., as well as a graduate research fellowship from the NIH Training Grant T32 GM-065103 to H.S-M. The authors thank Ryan Mehl for providing the plasmid for hCAII.

## Author contributions

H.S.-M., J.L.K. and D.K.S designed the work. H.S.-M. prepared all samples, carried out all experiments and performed data analysis. All authors contributed to data interpretation and paper preparation. J.L.K. and D.K.S. acquired the funding for this study.

## Competing interests

The authors declare the following competing financial interest(s): Although there is no direct conflict of interest, H.S-M, J.L.K and D.K.S. are inventors on a patent application on the use of mixed polymer brush surfaces for stabilizing immobilized enzymes.
