## [Peer Review File · Nature Communications]

Supra-Biological Performance of Immobilized Enzymes Enabled by Chaperone-like Specific Non-Covalent InteractionsReviewer #1 (Remarks to the Author):

The manuscript from Sanchez-Moran and colleagues ostensibly describes a characterization and optimization of lipase activity after immobilization on random copolymer brushes such that they exhibit increased activity and also at higher temperatures. This manuscript seems to be part of a series of publications from this group looking at the important issue of how to both stabilize and optimize the activity of enzymes when they are immobilized. In general, I found this manuscript to be very well written and interesting – but I do have some serious concerns about it. These include:

First. The data certainly shows that the lipase undergoes a concerted improvement in its activity and temperature profile in response to EGPMa. As best as I could understand, in Figure 3, the heat plots are each normalized to 1 as the highest value within that plot – but not between all the plots to the highest overall turnover measured? Figure S4 seems to support this. If this is so – then it is hard to really parse the data and compare the changes to each profile as a function of changing residues. Figure 3 should also show corresponding plots such as plot 2A or S4 and then the SI should also contain versions of those plots that are in linear and not a log plot. The data should also be presented in a manner similar to that shown in Figure S2 for comparison. That would allow the reader to better compare and appreciate what is going on.

Second. I am concerned about the overall interpretation. Lipases are known to have very complex binding and catalytic processes. Many of them have lids that cover the binding cleft, although this version does not. The binding cleft also undergoes a complex interaction with the substrate that is based on hydrophobicity and other factors. The Authors claim a chaperone-like interaction by the brushes – can it not just be that the mutations in conjunction with the surrounding environment of the brushes improve substrate binding interactions within the cleft or change the shape of the cleft? Same with the change in temperature? There is also a growing literature about how enzyme attachment to nanoparticles can alter and improve their activity.

Third. I am concerned about the generalizability of this strategy. I really think the Authors need to show this with another completely different enzyme to merit publication in this journal. How do we know that this is not just a result peculiar to this enzyme?

Minor points:

Please provide the size of the silica nanospheres that were used for immobilization.

Reviewer #2 (Remarks to the Author):

The manuscript “Supra-Biological Performance of Immobilized Enzymes Enabled by Chaperone-like Specific Non-Covalent Interactions” presents an interesting study to advance in the knowledge of functional-structural features relationship of solid-supported enzyme catalysts. Authors have taken the case of a well-studied lipase (*Bacillus subtilis* Lipase A (LipA)). Lipase is immobilised on a mode solid substrate, and the features of the surface carrier material have been finely tuned by controlling the density of aromatic moieties. This fine-tuning is aiming at modulating the surface-enzyme interaction upon immobilization and, therefore, affecting the catalytic properties of the resulting heterogeneous catalyst. The study aims to gain knowledge in the classic paradigm of the enzyme immobilisation field, which resides in the fact that structural changes upon immobilisation of an enzyme on a solid surface provoke critical modulation of catalytic properties. This paradigm is solidly founded on many experimental results documented in countless papers and summarized in many reviews and books. However, this paradigm mainly resides on experimental data from functional parameters evaluated, while the structural characterization of the immobilised enzymes is still very limited.

In this last sense, in the last years, the authoring group have published a seminal series of papers where new innovative methodologies such as single-molecule FRET provide insights into phenomena at the solid-enzyme interface and offer a new structure-function relationship of the heterogeneous enzyme catalysts. The present paper brings forward that understanding and provides a new level of fine-tuning of the structure-function relationship. For that the authors rely

on a previously developed strategy of the activation of the surface with random copolymer brushes that constitutes the active element of the surface to control immobilization. In this paper, authors focus on non-covalent interactions as the driving critical factor modulating the dynamics of the enzyme structure at the solid surface. This perspective is very interesting since the state of the art of enzyme immobilization focuses on the importance of strong multipoint covalent attachment of the immobilised enzymes as responsible of dramatic structural rigidification of the protein structure and the well-known activity-stability trade-off of the resulting catalyst. The possibility of tuning structure-function by weaker reversible interactions displays an additional powerful toolbox, very interesting for fine-tuning of the final properties of the catalyst, and it resembles in fact biological phenomena of enzyme performance modulation.

The manuscript is based on three pillars: fine-tuning of the surface of material support, characterization of the functional performance of immobilised lipase, and single molecule studies. The results display significant impact, and they are very interesting for the readership dealing with enzymes in solid-supported format. To impact a broader readership, guarantee excellent dissemination, and guarantee reproducibility, a few issues should be addressed before. The main two are affecting the functional characterization of the immobilised lipase and the practical implications of the findings in the operational performance of the catalyst.

Major issues: Preparation and functional characterization of the immobilised enzyme:

The activity and stability of the enzyme are characterized by using the hydrolysis of a model synthetic substrate (a butyrate ester), which is very common in the characterization of immobilised lipases where a chromogenic/fluorescent molecule (resorufin in this case) is released upon the hydrolysis and the extent of the reaction is followed. This a valid strategy but the experimental details should be enriched and probably new experiments performed to match the reach of the claims and the impact of the results, for example:

1. The experimental section (supporting information) should be enriched with details of the reaction set-up: reaction time, calibrations, temperature-dependent fluorescence, curves of spontaneous ester hydrolysis at high temperatures...
2. Data of among (mg, active sites number) per specific amount of solid support would be interesting to be reported in all the cases.
3. Different brushes are generated by fine-tuning with EGPMA/GMA. I wonder whether a more quantitative and spatial-defined characterization of the active groups can be made.
4. Since brushes have different EGPMA compositions, partition effects and the consequences on the local concentration of the substrates (resorufin butyrate and water) should be considered and the potential effect of reaction rate discussed.
5. Brushes contain GMA groups to ensure covalent attachment. I understand that it is very useful to avoid enzyme leaching during analysis and focus on enzyme-surface interaction, but the effect of the covalent interaction of the enzyme on the brushes should be considered and discussed: which amino acids are involved, different EGPMA doped brushes also have different degree of covalent attachment? How authors can be sure that in the double event of enzyme immobilisation: adsorption plus covalent attachment, the second one is identical in all the cases? Is the effect of the covalent attachment negligible for the activity/stability interplay or it has a synergistic effect with the adsorption?
6. The concentration of the substrate is very low, it is inferred that short reaction times are used to capture the initial reaction rate at low substrate conversions, but given the high impact on the relative activity, it is difficult to understand the substrate concentration time courses obtained.
7. Given the low substrate concentration used, the reviewer is curious about the dissection of the activity characterization into the two fundamental parameters of classic hyperbolic enzyme kinetics: K_{cat} (V_{max}) and K_m (at least admitting excess of water and simplification to the kinetic to a formal monosubstrate dependency). A more detailed kinetic characterisation could be very useful. For example, are both kinetic constants temperature dependent?
8. When the temperature is used as a fundamental intensive variable and activity is measured as initial reaction rates at short reaction times, every experiment is affected by at least two phenomena taking place: the effect of the temperature on the (substrate consumption) kinetics, and the effect of the temperature on the inactivation kinetics. Authors should comment on that, and the suggested more detailed kinetic analysis can help here. Please reconsider the use of terms

such as optimum temperature. Although these terms are classically used in kinetic studies in enzymology, the term can be misleading from other perspectives (e.g. process engineering)

Major issues: Practical implications of the results for the operational performance of the immobilised enzyme.

1. Following the previous discussion, a complete and broadly useful characterization of an enzyme catalyst normally implies using substrate consumption/product formation time courses at different operational conditions (in this case substrate concentration and temperature as a fundamental variable).

2. Figure 2d, although is not in reacting conditions, follows the operational perspective, and indeed, the data extracted from there in Figure 2e shows a complementary needed perspective. The impact of the fine-tuning is quantitatively different to the one based on activity relying on initial reaction rates. I find it very interesting to deepen there to gain a broader and deeper impact of the paper.

3. I find it interesting and curious whether the immobilised enzyme performance can be tested with other substrates and whether same effects are observed.

Other issues:

- Introduction and state of the art need to be updated with more recent papers and recent overviews of enzyme immobilisation.
- The first part of the results are introduction from the work of the authors, it should be integrated into the introduction.
- Based on the revision that the authors will performed, abstract, conclusions and the use of some words (supra-biological, chaperone-like, optimal, optimum...) should be reconsidered or better explained.

Reviewer #3 (Remarks to the Author):

Q.1 What are the noteworthy results?

A.1 Stabilization of enzymes using random copolymers via multiple interactions

Q.2 Will the work be of significance to the field and related fields? How does it compare to the established literature? If the work is not original, please provide relevant references.

A.2 There are many studies on the stabilization of enzymes using random copolymers. For example, Ting Xu group has worked on stabilizing MANY enzymes using random heteropolymers in toluene to make enzyme-polymer materials using polar, non-polar and charged groups grafted on chains (see Panganiban, B. et al. Random heteropolymers preserve protein function in foreign environments. *Science* 359, 1239–1243 (2018)). Her latest work with reference Z. Ruan, S. Li, A. Grigoropoulos, H. Amiri, S. L. Hilburg, H. Chen, I. Jayapurna, T. Jiang, Z. Gu, A. Alexander-Katz, C. Bustamante, H. Huang & T. Xu Population-based heteropolymer design to mimic protein mixtures. *Nature* 615, pages 251–258 (2023), has exploited the design of such random heteropolymers.

Actually, Ting Xu group has MANY papers trying to find the optimal composition of heteropolymers to stabilize enzymes and others have studied the mechanism (see Nguyen, T. D.; Qiao, B.; Olvera de la Cruz, M. Efficient encapsulation of proteins with random copolymers. *Proc. Natl. Acad. Sci. U. S. A.* 115, 6578– 6583 (2018)), and explored such ideas to form enzymes-polymer complexes (see Cardellini, A.; Jiménez-Ángeles, F.; Asinari, P.; Olvera de la Cruz, M. A Modeling-Based Design to Engineering Protein Hydrogels with Random Copolymers. *ACS Nano* 15, 16139– 16148 (2021)) and to decompose PET (see C. Waltmann, C. E. Mills, J. Wang, B. Qiao, J. M. Torkelson, D. Tullman-Ercek, and M. Olvera de la Cruz, "Functional Enzyme-Polymer Complexes" *PNAS* 119 (13), e2119509119 (2022))

It is not clear why all these reference are missing in this work.

Regarding the quality of the work: i) the work supports the conclusions and claims; ii) here are no flaws in the data analysis, interpretation and conclusions; iii) the methodology is sound and the work meets the expected standards in the field; iv) there is enough detail in the methods for the work to be reproduced.

Reviewer #1:

The manuscript from Sanchez-Moran and colleagues ostensibly describes a characterization and optimization of lipase activity after immobilization on random copolymer brushes such that they exhibit increased activity and also at higher temperatures. This manuscript seems to be part of a series of publications from this group looking at the important issue of how to both stabilize and optimize the activity of enzymes when they are immobilized. In general, I found this manuscript to be very well written and interesting – but I do have some serious concerns about it. These include:

We thank the reviewer for their favorable comments and overall interest in our manuscript.

First. The data certainly shows that the lipase undergoes a concerted improvement in its activity and temperature profile in response to EGPMA. As best as I could understand, in Figure 3, the heat plots are each normalized to 1 as the highest value within that plot – but not between all the plots to the highest overall turnover measured? Figure S4 seems to support this. If this is so – then it is hard to really parse the data and compare the changes to each profile as a function of changing residues. Figure 3 should also show corresponding plots such as plot 2A or S4 and then the SI should also contain versions of those plots that are in linear and not a log plot. The data should also be presented in a manner similar to that shown in Figure S2 for comparison. That would allow the reader to better compare and appreciate what is going on.

The reviewer is correct that the heatmaps in **Figure 3** were indeed each normalized to the highest activity for that variant. We feel this is a convenient way to visualize how the different variants were optimized by different support compositions, which was the main point of emphasis (as different mutants are expected to have different absolute activities). Nevertheless, we agree it would also be appropriate to show the raw activity data such that differences in the activity of the variants can be clearly observed. Given this, we have modified **Figure 3** in the revised manuscript to show the raw activity data along with the normalized heatmaps. Additionally, versions of these figures in which the activity data is plotted on a linear scale have been included in the supporting information of the revised manuscript (see **Supplementary Figure S5**). While the variants do differ in absolute activity, this does not alter the main conclusion from these data, which involves the correlation between aromatic residues and aromatic composition of optimally stabilizing brushes. In line with this change, we have also added the corresponding linear plots for **Figure 2a** (**Supplementary Figure S2**) and for **Figure 6a** (**Supplementary Figure S12**). We thank the reviewer for this suggestion and believe the changes improve the clarity and overall quality of the manuscript.

Second. I am concerned about the overall interpretation. Lipases are known to have very complex binding and catalytic processes. Many of them have lids that cover the binding cleft, although this version does not. The binding cleft also undergoes a complex interaction with the substrate that is based on hydrophobicity and other factors. The Authors claim a chaperone-like interaction by the brushes – can it not just be that the mutations in conjunction with the surrounding environment of the brushes improve substrate binding interactions within the cleft or change the shape of the cleft? Same with the change in temperature? There is also a growing literature about how enzyme attachment to nanoparticles can alter and improve their activity.

While the SM-FRET experiments directly demonstrate that properly tuned supports promote the refolding of LipA (i.e. the chaperone effect), the reviewer correctly notes that the supports could also cause an additional effect involving substrate binding. To address this, we have performed additional experiments to probe potential changes in substrate binding associated with the support composition.

Specifically, to test whether the composition of the support influenced substrate binding, the K_m of immobilized LipA was measured on each of the supports. For Michaelis-Menten analysis, we used the wild-type form of LipA and found the value of K_m for immobilized LipA_{WT} was similar on all surface compositions at both 20 °C and 45 °C. Hence, the differences in catalytic activity of LipA_{WT} on different supports could be attributed to other effects, such as differences in thermal stabilization. A brief discussion of the measurement K_m for immobilized wild-type LipA was added on page 10 in the revised manuscript and the values of K_m at 20 and 45°C have also been included in the supporting information (see **Supplementary Table S3**).

Additionally, to determine the impact of the mutations on substrate binding, we measured the K_m for the soluble form of each of the variants and showed that the values of K_m for LipA_{WT}, LipA_{ΔSA-50%}, and LipA_{Δburied-100%} were similar whereas the K_m for LipA_{ΔSA-100%} was higher than the others. The latter was not surprising since Y161A, one of the mutations in LipA_{ΔSA-100%}, is close to the active site and could affect substrate binding. An explanation of Michaelis-Menten analysis for the variants has been added on page 12 in the revised manuscript with the values of K_m now reported in **Supplementary Table S5**.

Additionally, the reviewer raises an important point related to the general mechanism of lipases involving the amphiphilic lid that often covers the active site. In fact, the activation of lipases upon immobilization is often attributed to the opening of this lid domain due to interactions (namely hydrophobic) between the lid and immobilization support. However, as the reviewer goes on to note, the lipase used in this work (*Bacillus subtilis* Lipase A) is unique in that it does not have a lid like many other lipases. Given this, the impact of the brush surfaces on the activation and stabilization of the lipase variants is unrelated to changes in the conformation and/or dynamics of the lid domain. A brief discussion of this point along with references on the structure of *Bacillus subtilis* Lipase A have been added in the revised manuscript (on page 4).

Finally, we note that while changes in substrate binding do not appear to explain the differences in activity associated with different support compositions, the chaperone-like effect we describe was directly measured using SM-FRET with LipA_{WT}. This effect was inferred from the enzyme's folding and unfolding dynamics on different supports. In the SM-FRET experiments, the labeled sites in the enzyme were remote and on the opposite face of the enzyme from the active site (updated in **Figure 4**). As such, changes in FRET are reflective of gross dynamic changes in the tertiary structure, which was consistent with ensemble chemical denaturation experiments (**Supplementary Figure S8**). Thus, the chaperone effect refers to the fact that optimized brush compositions facilitated the refolding of transiently refolded enzyme molecules, leading to an increase in the time-averaged folded fraction, which was consistent with increased enzyme activity when immobilized on those optimized support compositions. A short note explaining this point has been added on page 14 in the revised manuscript.

Third. I am concerned about the generalizability of this strategy. I really think the Authors need to show this with another completely different enzyme to merit publication in this journal. How do we know that this is not just a result peculiar to this enzyme?

This is an excellent suggestion, and we agree that the use of a completely unrelated second enzyme would help demonstrate the general utility of this approach. In response to this suggestion, we have performed experiments using human carbonic anhydrase II (hCAII). hCAII is structurally unrelated to LipA, and catalyzes a different reaction and was therefore assayed using a different substrate, which was also fluorogenic. Consistent with the results for LipA, a large stabilizing effect was observed for hCAII with an optimum of 10% EGPMA in the brush layer. In particular, upon immobilization on SBMA brushes mixed with EGPMA, hCAII exhibited increasing activity as a function of temperature all the way up to 90°C. For comparison, soluble hCAII is completely inactive at 90°C and begins to lose activity at 70°C. Such stabilization of hCAII is unprecedented to our knowledge and has significant implications for potential applications of hCAII in carbon capture. These data has been added to the revised manuscript along with a discussion of these findings and their potential industrial implications (on pages 21-23). Ultimately, the inclusion of this data significantly strengthens the manuscript in our view and will enhance the impact of our findings to the broader community, for which we thank the reviewer.

Minor points:

Please provide the size of the silica nanospheres that were used for immobilization.

We thank the reviewer for pointing out this detail that was missing. In the revised manuscript, the hydrodynamic diameter of the silica nanospheres (411 nm) was highlighted on page 6 in the results and discussion section.

Reviewer #2:

The manuscript “Supra-Biological Performance of Immobilized Enzymes Enabled by Chaperone-like Specific Non-Covalent Interactions” presents an interesting study to advance in the knowledge of functional-structural features relationship of solid-supported enzyme catalysts. Authors have taken the case of a well-studied lipase (Bacillus subtilis Lipase A (LipA)). Lipase is immobilised on a mode solid substrate, and the features of the surface carrier material have been finely tuned by controlling the density of aromatic moieties. This fine-tuning is aiming at modulating the surface-enzyme interaction upon immobilization and, therefore, affecting the catalytic properties of the resulting heterogeneous catalyst. The study aims to gain knowledge in the classic paradigm of the enzyme immobilisation field, which resides in the fact that structural changes upon immobilisation of an enzyme on a solid surface provoke critical modulation of catalytic properties. This paradigm is solidly founded on many experimental results documented in countless papers and summarized in many reviews and books. However, this paradigm mainly resides on experimental data from functional parameters evaluated, while the structural characterization of the immobilised enzymes is still very limited.

In this last sense, in the last years, the authoring group have published a seminal series of papers where new innovative methodologies such as single-molecule FRET provide insights into

phenomena at the solid-enzyme interface and offer a new structure-function relationship of the heterogeneous enzyme catalysts. The present paper brings forward that understanding and provides a new level of fine-tuning of the structure-function relationship. For that the authors rely on a previously developed strategy of the activation of the surface with random copolymer brushes that constitutes the active element of the surface to control immobilization. In this paper, authors focus on non-covalent interactions as the driving critical factor modulating the dynamics of the enzyme structure at the solid surface. This perspective is very interesting since the state of the art of enzyme immobilization focuses on the importance of strong multipoint covalent attachment of the immobilised enzymes as responsible of dramatic structural rigidification of the protein structure and the well-known activity-stability trade-off of the resulting catalyst. The possibility of tuning structure-function by weaker reversible interactions displays an additional powerful toolbox, very interesting for fine-tuning of the final properties of the catalyst, and it resembles in fact biological phenomena of enzyme performance modulation.

The manuscript is based on three pillars: fine-tuning of the surface of material support, characterization of the functional performance of immobilised lipase, and single molecule studies. The results display significant impact, and they are very interesting for the readership dealing with enzymes in solid-supported format. To impact a broader readership, guarantee excellent dissemination, and guarantee reproducibility, a few issues should be addressed before. The main two are affecting the functional characterization of the immobilised lipase and the practical implications of the findings in the operational performance of the catalyst.

We thank the reviewer for noting that our work reflects significant impact and will be of broad interest to the biocatalysis community.

Major issues: Preparation and functional characterization of the immobilised enzyme:

The activity and stability of the enzyme are characterized by using the hydrolysis of a model synthetic substrate (a butyrate ester), which is very common in the characterization of immobilised lipases where a chromogenic/fluorescent molecule (resorufin in this case) is released upon the hydrolysis and the extent of the reaction is followed. This a valid strategy but the experimental details should be enriched and probably new experiments performed to match the reach of the claims and the impact of the results, for example:

1. The experimental section (supporting information) should be enriched with details of the reaction set-up: reaction time, calibrations, temperature-dependent fluorescence, curves of spontaneous ester hydrolysis at high temperatures...

As the reviewer suggests, additional details on the methods used for assaying the activity of LipA have been added in the revised manuscript. Specifically, we have included a detailed description of the length of time over which the product of hydrolysis was followed in the methods section (on pages 30-31) in the revised manuscript. As part of this, we have also specified the frequency that fluorescence that measurements were recorded to go along with other details such as substrate concentration and buffer conditions. Additionally, plots for the calibration curve of fluorescence versus resorufin concentration and the background hydrolysis of the substrate as a function of

temperature have been included in the supporting information. Moreover, we also confirmed that the fluorescence of the hydrolysis product was stable between 20 – 90°C, the temperature range used in this work, and have noted this in the manuscript. Finally, we have also added similar experimental details for carbonic anhydrase, which was used to further demonstrate the utility of approach as explained in the above response to reviewer #1. Notably, for carbonic anhydrase, we used 4-methylumbelliferyl acetate as a substrate, which is converted to the fluorescent product 4-methylumbelliferone. The details related to substrate hydrolysis and temperature-dependent fluorescence for 4-methylumbelliferyl acetate have been added in the methods section in the revised manuscript on page 31.

2. Data of among (mg, active sites number) per specific amount of solid support would be interesting to be reported in all the cases.

We agree with the reviewer that it would be useful to report the number of active sites per mass of each of the lipase variants used in our experiments. To address this, we have estimated the number of active sites using the enzyme mass per mass of support combined with the folded fractions determined from single-molecule FRET experiments. These data for wild-type LipA were added in **Supplementary Table S2** and discussed on page 7. The same data for the LipA mutants and hCAII are included in **Supplementary Tables S6** and **S8**, and briefly discussed on pages 11 and 21 in the revised manuscript.

3. Different brushes are generated by fine-tuning with EGPMA/GMA. I wonder whether a more quantitative and spatial-defined characterization of the active groups can be made.

The reviewer raises an interesting question about the spatial distribution of epoxide reactive groups (i.e., GMA) within the brush layer. We agree that it is interesting to consider a more quantitative picture of the GMA distribution in the polymer brushes, but this information is not readily accessible using experimental methods. In some of our previous work, we observed that the classical assumption that all methacrylate monomers have equal reactivity held true by using Fourier transform infrared spectroscopy (doi.org/10.1021/acsabm.9b00815), and while we tried to verify the same in this study by using diffusive reflectance infrared Fourier transform spectroscopy, the small epoxide signal overlapped with other signals, making it impossible to quantify its content. However, despite equal reactivity, we believe it is interesting to consider how changing the SBMA/EGPMA ratio could alter the density of GMA moieties. This is because despite the well-defined molar ratios, there are subtle differences in molecular volume for all three monomers. Our calculations suggest that changes in GMA density as a function of EGPMA were negligible, with an approximate volume of 2.5% of the total brush for all compositions. We have added a short discussion discussed on page 6 in the revised manuscript.

4. Since brushes have different EGPMA compositions, partition effects and the consequences on the local concentration of the substrates (resorufin butyrate and water) should be considered and the potential effect of reaction rate discussed.

The reviewer brings up an important topic, and we have performed additional experiments to probe the partitioning as a function of support composition. Using resorufin as a fluorescent reporter, we characterized the partitioning into supports with different EGPMA content, and measured

fluorescence retention of supports at 20, 45 and 70°C. While resorufin lacks the butyrate moiety of the substrate, overall, it is a structurally similar analogue. Specifically, particles coated with SBMA/EGPMA copolymer were incubated with 50 µM resorufin for 20 minutes at room temperature and subsequently subjected to 3 rounds of washing with buffer at 20, 45 or 70 °C. After washing, the remaining fluorescence of the support particles was measured, providing information about differential resorufin partitioning as a function of EGPMA content. Particles without GMA were used to avoid covalent coupling of resorufin to the epoxide moiety of GMA.

We found that for each temperature, there were no measurable differences in resorufin uptake as a function of EGPMA content in the brush layer. Interestingly, we also observed that the partitioning effect greatly decreased with increasing temperature. This suggested that the catalytic enhancements at high temperature purely reflected an increase in the rate of enzymatic catalysis and thermal stabilization, and not a hypothetical increase of local concentration of substrate, which could artificially alter the observed rate. To clarify this, we have added a brief discussion of this on page 10 in the revised manuscript while the results of this experiment have also been added in the supporting information (see **Supplementary Figure S3**).

Additionally, the reviewer also brings up the need of water as a substrate for lipase-catalyzed hydrolysis, which is often overlooked and could be especially critical in environments which are highly depleted from water. We believe that in this case, LipA-catalyzed hydrolysis reactions could be approximated as a single substrate reaction, given that SBMA-containing polymer brushes are well solvated (doi/10.1021/ja5065334), which creates a substantial excess of water at the brush interface. To clarify this, we have added a sentence on page 6 in the revised manuscript explaining this point.

5. Brushes contain GMA groups to ensure covalent attachment. I understand that it is very useful to avoid enzyme leaching during analysis and focus on enzyme-surface interaction, but the effect of the covalent interaction of the enzyme on the brushes should be considered and discussed: which amino acids are involved, different EGPMA doped brushes also have different degree of covalent attachment? How authors can be sure that in the double event of enzyme immobilisation: adsorption plus covalent attachment, the second one is identical in all the cases? Is the effect of the covalent attachment negligible for the activity/stability interplay or it has a synergistic effect with the adsorption?

The reviewer raises several interesting questions related to the immobilization of LipA to the mixed EGPMA/SBMA/GMA brushes. The process of covalent immobilization through GMA can indeed involve reaction with numerous sites on the surface of enzymes, including the N-terminus, lysines, cysteines, tyrosines, and histidines. Given the large number of these residues present on LipA's surface, each enzyme molecule may potentially react with more than one GMA monomer. To predict the most reactive residues in LipA, we analyzed the pKa and solvent accessibility for all possible reactive residues, including the N-terminus. Based on pKa and solvent accessibility, we were able to classify each residue as fast reacting, slow reacting, or non-reactive. The results of this analysis have been added in the supporting information (see **Supplementary Table S1**) and are described on page 6 in the revised manuscript.

Furthermore, although the enzyme may interact non-covalently with the brush, SBMA-based brushes are typically classified as “superlow” protein adsorbing coating materials, hence, our materials are highly resistant to protein adsorption given the low content of EGPMA. Despite this, we rule out any potential adsorption by extensively washing particles upon immobilization to remove non-covalently adsorbed enzyme. A phrase has been added in the results section on page 6, as well as a phrase in the methods (on page 30).

Lastly, the reviewer asks about the effect of immobilization on the intrinsic activity of the enzyme (i.e., if it activates or deactivates by the mere immobilization). In **Figure 2a**, we can observe that LipA’s intrinsic activity increases by immobilization, hence, there is a synergistic activation and stabilization effect as the reviewer notes. Conversely, hCAII, is intrinsically deactivated by immobilization (**Figure 6**), hence, we observe an activity/stability trade-off. Nonetheless, despite deactivation by immobilization, the catalytic rates of hCAII at high temperatures are surpassed by 9-fold, which highlights the importance of thermal stabilization for the enhancement of catalytic rates. For LipA, we have added a phrase noting the synergistic effect on page 8, and for hCAII, we have added a phrase on page 22 noting the observed trade-off.

6. The concentration of the substrate is very low, it is inferred that short reaction times are used to capture the initial reaction rate at low substrate conversions, but given the high impact on the relative activity, it is difficult to understand the substrate concentration time courses obtained.

As pointed out by the reviewer, the concentration of substrate used for activity assays was rather low. This was due to the low solubility of the substrate in buffer. It is important to note that the substrate concentration was higher than the K_m for LipA in all cases, based on the values of the apparent K_m values that were measured for soluble and immobilized LipA (including the soluble LipA variants). Also, the concentration of substrate was sufficient to measure the initial reaction rate (as noted by the reviewer) of enzyme that had been subjected to various thermal stresses for different amounts of time. We have clarified the workflow for this experiment in the Methods section (page 30).

7. Given the low substrate concentration used, the reviewer is curious about the dissection of the activity characterization into the two fundamental parameters of classic hyperbolic enzyme kinetics: K_{cat} (V_{max}) and K_m (at least admitting excess of water and simplification to the kinetic to a formal monosubstrate dependency). A more detailed kinetic characterisation could be very useful. For example, are both kinetic constants temperature dependent?

As suggested by the reviewer, we measured the k_{cat} and K_m values for LipA on each of the supports at 20°C and 45°C. For determination of k_{cat} at both temperatures, we used the apparent folded fractions from SM-FRET experiments to estimate the amount of active enzyme for each condition. These experiments revealed that substrate binding (K_m) was independent of EGPMA content for both temperatures. Interestingly, K_m increased with higher temperature, which could be explained by the substrate partitioning effect reported in **Supplementary Figure S3**, which would artificially increase substrate concentration in the brush layer at low temperatures. Therefore, changes in substrate binding did not explain the increase in enzyme activity at higher temperature. Instead, our results suggest that the increase in enzyme activity may be explained by differences in k_{cat} along with variations in the amount of folded LipA. Indeed, 5% EGPMA support being the most

active in **Figure 2a** may be explained by a dual effect of 5% EGPMA having both the highest k_{cat} and preserving the most folded LipA. The results of this analysis have been added in the supporting information (see **Supplementary Table S3**) while a brief discussion has been added on page 10 in the revised manuscript. We thank the reviewer for this question, which we believe helps deepen the understanding of the nature of catalytic enhancement for immobilized LipA.

8. When the temperature is used as a fundamental intensive variable and activity is measured as initial reaction rates at short reaction times, every experiment is affected by at least two phenomena taking place: the effect of the temperature on the (substrate consumption) kinetics, and the effect of the temperature on the inactivation kinetics. Authors should comment on that, and the suggested more detailed kinetic analysis can help here. Please reconsider the use of terms such as optimum temperature. Although these terms are classically used in kinetic studies in enzymology, the term can be misleading from other perspectives (e.g. process engineering)

As noted by the reviewer, temperature may significantly impact background hydrolysis of substrate as well as the kinetics of inactivation of the immobilized enzyme. To address this, we measured the background hydrolysis of resorufin butyrate as a function of temperature as described in our response above. In addition to including this data in the revised manuscript, we have included a brief discussion on how increasing temperature can enhance reaction rates and also increase enzyme unfolding, leading to inactivation (see page 10 in the revised manuscript).

Furthermore, while we agree that the terms used may have different means in other contexts, we in fact view this work as aligning more closely to classical enzymology than process engineering. Consistent with this, we feel that the use of the terms optimum temperature and optimum rate are appropriate in this context and important for connecting to similar work. Nonetheless, we have carefully reviewed the manuscript to ensure we have clearly defined the meaning of these terms as well as others that may have alternative meanings to avoid any confusion.

Major issues: Practical implications of the results for the operational performance of the immobilised enzyme.

1. Following the previous discussion, a complete and broadly useful characterization of an enzyme catalyst normally implies using substrate consumption/product formation time courses at different operational conditions (in this case substrate concentration and temperature as a fundamental variable).

While we agree that it can be practically useful to measure enzyme progress curves under conditions that mimic industrial processes, we wish to stress that the goal of this work was not to demonstrate a process, but instead investigate the molecular basis for a novel enzyme stabilization strategy via new materials. As such, the main focus of the paper is to clearly demonstrate the effect of the brushes on the activity and stability of LipA, while explaining the mechanism for the observed effects. We note that this is already a long and comprehensive manuscript, incorporating thorough characterization of reactions employing multiple enzymes and mutants and a large range of support compositions, as well as single-molecule studies that support mechanistic interpretations. As such, we feel that the detailed studies of potential industrial processes would be better suited to a separate follow-up paper.

2. Figure 2d, although is not in reacting conditions, follows the operational perspective, and indeed, the data extracted from there in Figure 2e shows a complementary needed perspective. The impact of the fine-tuning is quantitatively different to the one based on activity relying on initial reaction rates. I find it very interesting to deepen there to gain a broader and deeper impact of the paper.

We appreciate this suggestion and have added a concise discussion about the interplay between initial activity and stabilization over time. We believe that this discussion provides a more complete picture about the ways in which the supports may improve performance under operational conditions. We have added a brief discussion on page 11 discussing this impact.

3. I find it interesting and curious whether the immobilised enzyme performance can be tested with other substrates and whether same effects are observed.

Although we did not perform tests using other substrates for LipA, similar effects were observed with carbonic anhydrase as indicated above in response to the comment from reviewer 1 about using other enzymes. Notably, by testing this approach with carbonic anhydrase, we inherently showed that this approach to stabilizing enzymes was independent of the enzyme and substrate used. While certain substrates may interact with the brush layer differently, we have not observed any evidence of this in this work. This point was highlighted on page 21 in the discussion of the results in the revised manuscript.

Other issues:

- Introduction and state of the art need to be updated with more recent papers and recent overviews of enzyme immobilisation.

As suggested by the reviewer, we have updated the discussion and references in the introduction to reflect more recent work in the field of enzyme immobilization. In particular, we highlighted several other studies where polymer brushes have been used as immobilization supports for enzymes. Additionally, we have also added several references on the use of complex heterogeneous polymers for stabilizing enzymes. We believe these changes significantly improve the depth and scope of the introduction section.

- The first part of the results are introduction from the work of the authors, it should be integrated into the introduction.

We have restructured the introduction and initial paragraph of the results section, so that previous background is indeed comprised in the introduction.

- Based on the revision that the authors will perform, abstract, conclusions and the use of some words (supra-biological, chaperone-like, optimal, optimum...) should be reconsidered or better explained.

We thank the reviewer for pointing out these potential sources of confusion. We have carefully reviewed the manuscript to ensure that the terms “chaperone-like” and “supra-biological” are fully

explained in the revised manuscript (on pages 4 and 5). As explained above, we have kept the words “optimal/optimum” throughout the manuscript for referring to optimal activity and temperature, which we believe are the accepted terms used in an enzymology context.

Reviewer #3:

There are many studies on the stabilization of enzymes using random copolymers. For example, Ting Xu group has worked on stabilizing MANY enzymes using random heteropolymers in toluene to make enzyme-polymer materials using polar, non-polar and charged groups grafted on chains (see Panganiban, B. et al. Random heteropolymers preserve protein function in foreign environments. Science 359, 1239–1243 (2018)). Her latest work with reference Z. Ruan, S. Li, A. Grigoropoulos, H. Amiri, S. L. Hilburg, H. Chen, I. Jayapurna, T. Jiang, Z. Gu, A. Alexander-Katz, C. Bustamante, H. Huang & T. Xu Population-based heteropolymer design to mimic protein mixtures. Nature 615, pages 251–258 (2023), has exploited the design of such random heteropolymers.

Actually, Ting Xu group has MANY papers trying to find the optimal composition of heteropolymers to stabilize enzymes and others have studied the mechanism (see Nguyen, T. D.; Qiao, B.; Olvera de la Cruz, M. Efficient encapsulation of proteins with random copolymers. Proc. Natl. Acad. Sci. U. S. A. 115, 6578– 6583 (2018)), and explored such ideas to form enzymes-polymer complexes (see Cardellini, A.; Jiménez-Ángeles, F.; Asinari, P.; Olvera de la Cruz, M. A Modeling-Based Design to Engineering Protein Hydrogels with Random Copolymers. ACS Nano 15, 16139– 16148 (2021)) and to decompose PET (see C. Waltmann, C. E. Mills, J. Wang, B. Qiao, J. M. Torkelson, D. Tullman-Ercek, and M. Olvera de la Cruz, "Functional Enzyme-Polymer Complexes" PNAS 119 (13), e2119509119 (2022))

It is not clear why all these references are missing in this work.

We thank the reviewer for highlighting these works and have included citations to them in the introduction of the revised manuscript (page 4). In addition to these citations, additional discussion on these works has also been included in the revised manuscript. We agree that the inclusion of this discussion and is both relevant and important and recognize that this should have been included in the initial version of the manuscript.

Regarding the quality of the work: i) the work supports the conclusions and claims; ii) there are no flaws in the data analysis, interpretation and conclusions; iii) the methodology is sound and the work meets the expected standards in the field; iv) there is enough detail in the methods for the work to be reproduced.

We thank the reviewer for their appreciation of the rigor and level of detail in the manuscript, and their overall favorable comments in general.

Reviewer #1 (Remarks to the Author):

The Authors have made a concerted effort towards revising this manuscript. I found that the manuscript is substantially improved. In particular, the Authors have addressed all the issues that I raised during the review and addressed each of the questions with appropriate additional experiments or responses. They also annotated and updated the manuscript accordingly. The manuscript is now acceptable.

Reviewer #2 (Remarks to the Author):

Authors have performed an extensive, detailed and rigorous revision of the manuscript. New discussion and explanations were added where requested, and new experimental data were included, including the application of the methodology to a second enzyme where similar conclusions were drawn. It was a pleasure to read the new version of the manuscript and the rebuttal letter, I enjoyed the discussion with the authors about insights into fine-tuned stabilization of enzymes by immobilization. At this point, I can just congratulate the authors and wish nice dissemination of the work. I just have very few minor points to be considered:

- In the abstract: Contextualize the data of 50-fold enhancement. Enhancement regarding what?
- In the introduction: I fully agree that new materials with newly tuned surfaces are critical to obtaining better solid-supported enzymes and under the state of the art our knowledge of the functional properties of immobilized enzymes given by the interplay of surface, material, and chemistry...is limited and not sufficient. But I am not very happy with the sentence "Unfortunately, many of these materials are inherently damaging to enzymes upon immobilization". I would nuance and polish it.
- In conclusions: I agree with the authors that this is not a paper in the context of process engineering but on fundamental understanding of the enzymology of immobilized enzymes, but still the reach and potential of the work displayed here will be of interest to bioprocess engineers. Some conclusions or outlook in that direction can be briefly added.
- In the page 10 the authors included a discussion of the kinetic parameters calculated. I agree with the conclusions and the basics. But I would suggest a final polishing of the text. It is not immediately clear the position regarding the variation of substrate binding (K_m) at high temperatures. The second sentence can be confusing compared with the upcoming discussion.

Reviewer #1 (Remarks to the Author):

The Authors have made a concerted effort towards revising this manuscript. I found that the manuscript is substantially improved. In particular, the Authors have addressed all the issues that I raised during the review and addressed each of the questions with appropriate additional experiments or responses. They also annotated and updated the manuscript accordingly. The manuscript is now acceptable.

We thank the reviewer for their appreciation of the revisions to the original manuscript and for recommending the manuscript is now acceptable for publication.

Reviewer #2 (Remarks to the Author):

Authors have performed an extensive, detailed and rigorous revision of the manuscript. New discussion and explanations were added where requested, and new experimental data were included, including the application of the methodology to a second enzyme where similar conclusions were drawn. It was a pleasure to read the new version of the manuscript and the rebuttal letter, I enjoyed the discussion with the authors about insights into fine-tuned stabilization of enzymes by immobilization. At this point, I can just congratulate the authors and wish nice dissemination of the work. I just have very few minor points to be considered:

We are grateful to the reviewer for their favorable comments towards the revised manuscript and have addressed the minor points raised below.

- In the abstract: Contextualize the data of 50-fold enhancement. Enhancement regarding what?

We thank the reviewer for pointing this out the lack of clarity around this statement and have revised this to specify that the fold enhancement refers to enzymatic activity.

- In the introduction: I fully agree that new materials with newly tuned surfaces are critical to obtaining better solid-supported enzymes and under the state of the art our knowledge of the functional properties of immobilized enzymes given by the interplay of surface, material, and chemistry...is limited and not sufficient. But I am not very happy with the sentence "Unfortunately, many of these materials are inherently damaging to enzymes upon immobilization". I would nuance and polish it.

As suggested by the reviewer, we have revised this sentence, which now reads:

"However, enzymes are widely inactivated upon immobilization on such materials due to denaturation as a result of enzyme-material interactions and/or blockage of the enzyme's active site, which in turn leads to a reduction in overall catalytic performance²⁴⁻²⁶.

- In conclusions: I agree with the authors that this is not a paper in the context of process engineering but on fundamental understanding of the enzymology of immobilized enzymes, but

still the reach and potential of the work displayed here will be of interest to bioprocess engineers. Some conclusions or outlook in that direction can be briefly added.

We agree with the reviewer that this work may be of interest to bioprocess engineers and have added the following sentence in the conclusions:

“By promoting the stabilization of enzymes in such environments, our findings may furthermore have significant impact on bioprocess optimization and biosystems reaction engineering.”

- In the page 10 the authors included a discussion of the kinetic parameters calculated. I agree with the conclusions and the basics. But I would suggest a final polishing of the text. It is not immediately clear the position regarding the variation of substrate binding (K_m) at high temperatures. The second sentence can be confusing compared with the upcoming discussion.

We agree that this paragraph could be polished to clearly convey that the catalytic effects that are observed are due to changes in k_{cat} and active site preservation, and not due to differences in K_m . We also aimed to clarify the apparent dependence of K_m with temperature, which we attribute to substrate binding at different temperatures. Therefore, we have restructured the paragraph and added several sentences in page 10.